# Lithium hexamethyldisilazide initiated superfast ring opening polymerization of alpha-amino acid N-carboxyanhydrides

Yueming Wu [1], Danfeng Zhang [1], Pengcheng Ma[1], Ruiyi Zhou[1], Lei Hua[2] & Runhui Liu [1]

Polypeptides have broad applications and can be prepared via ring-opening polymerization of $\alpha$-amino acid N-carboxyanhydrides (NCAs). Conventional initiators, such as primary amines, give slow NCA polymerization, which requires multiple days to reach completion and can result in substantial side reactions, especially for very reactive NCAs. Moreover, current NCA polymerizations are very sensitive to moisture and must typically be conducted in a glove box. Here we show that lithium hexamethyldisilazide (LiHMDS) initiates an extremely rapid NCA polymerization process that is completed within minutes or hours and can be conducted in an open vessel. Polypeptides with variable chain length (DP = 20–1294) and narrow molecular weight distribution (Mw/Mn = 1.08–1.28) were readily prepared with this approach. Mechanistic studies support an anionic ring opening polymerization mechanism. This living NCA polymerization method allowed rapid synthesis of polypeptide libraries for high-throughput functional screening.

[1] State Key Laboratory of Bioreactor Engineering, Key Laboratory for Ultrafine Materials of Ministry of Education, Research Center for Biomedical Materials of Ministry of Education, School of Materials Science and Engineering, East China University of Science and Technology, 200237 Shanghai, China. [2] Research Center of Analysis and Test, East China University of Science and Technology, 200237 Shanghai, China. Correspondence and requests for materials should be addressed to R.L. (email: rliu@ecust.edu.cn)

Synthetic polypeptides are biocompatible and have a broad range of applications including protein/peptide mimicking[1–4], antimicrobial agents and materials[5–14], drug, and gene delivery[15–18], tissue engineering and other biomaterial applications[19–25]. Polypeptides can be synthesized from the polymerization of α-amino acid N-carboxyanhydride (NCA), the Leuchs' anhydride discovered in 1906 by Hermann Leuchs[26–28]. Since that time, alcohols, amines and related compounds have been used to initiate NCA polymerization; primary amines have been the most common initiators[29–32]. In recent years, continues studies on NCA polymerization were reported by using transition metal complexes[33–35], hexamethyldisilazane and trimethylsilyl derivatives[36–38], Al-Schiff base complexes[39], rare earth metal complexes[40], ammonium salts[41–43], and amine-borane Lewis pairs as the initiator[44]. Other efforts have included the use of a thiourea organocatalyst[45], and the exploration of modified reaction conditions such as low temperature[46,47], high vacuum[48–50], and nitrogen flow[51]. Despite these innovations, NCA polymerization is still challenging due to its high sensitive to moisture and generally need to operate in a glove box.

We report the discovery that lithium hexamethyldisilazide (LiHMDS) initiates very rapid ring-opening NCA polymerization to generate polypeptides with low dispersity, over a wide range of average chain length. We show that LiHMDS-initiated NCA polymerization can be conducted on the benchtop, in an open vessel, which facilitates parallel synthesis (for library generation) and large-scale synthesis of specific materials. The merits of our method are illustrated via the preparation and screening of polypeptide libraries for antibacterial activity. We identify potent and broad-spectrum polymers that target antibiotic-resistant strains of *Pseudomonas aeruginosa* (*P. aeruginosa*) and methicillin-resistant *Staphylococcus aureus* (MRSA).

## Results

**Reaction rates of NCA polymerization initiated by different initiators**. LiHMDS has previously been used to initiate the anionic ring-opening polymerization of β-lactams to afford nylon-3 polymers that display a variety of biological applications[19,52–59]. We therefore wondered whether LiHMDS would be effective in deprotonating NH in the NCA ring and initiate NCA polymerization (Fig. 1a). Two widely-used NCAs were employed, those derived from γ-benzyl-L-glutamate NCA (BLG NCA) and from Nε-tert-butyloxycarbonyl-D,L-lysine NCA (Boc-D,L-Lys NCA) (see NCA synthesis details in Supplementary Methods and Supplementary Figs. 8–22), so that we could compare our findings to those of others. Many previous studies have employed DMF or THF as the solvent for NCA polymerization; our preliminary studies revealed that THF is superior because the NCAs are more stable in this solvent than in DMF to minimize side reactions possibly associated with impurities in solvents such as water and chloride ions (see NCA stability studies in Supplementary Discussion and Supplementary Fig. 1)[60–62].

Surprisingly, we observed that LiHMDS causes superfast polymerization of each NCA. For reactions involving 20:1 NCA:LiHMDS, both BLG NCA and Boc-D,L-Lys NCA were consumed very quickly, and the reactions were complete within 10 min and 15 min, respectively (Fig. 1). For reactions involving 100:1 or 500:1 NCA:LiHMDS, BLG NCA was consumed within 2 h, whereas Boc-D,L-Lys NCA required 7 or 21 h for complete consumption. We used the results summarized above as a basis for comparing our method with NCA polymerization methods described previously. Hexylamine, hexamethyldisilazane and bipyNi(COD) (Supplementary Table 1 and Supplementary Fig. 23 for NMR) were employed for these comparisons. With both hexylamine and hexamethyldisilazane, very little polymerization was detected at any of the time points we monitored for either NCA starting material (Fig. 1). Excellent work by Deming et al. has established that bipyNi(COD) is an effective catalyst for NCA polymerization[34], and our observations reinforce the advantages of the Ni-catalyzed process relative to polymerizations initiated with an amine. However, our data clearly establish the superiority of LiHMDS-initiated polymerizations, particularly those conducted with high NCA:initiator ratios (i.e., for synthesis of polymers with high molecular weight).

**Synthesis and characterization of poly-BLG and poly-Boc-D,L-Lys**. Above observation on LiHMDS-initiated superfast NCA polymerization encouraged us to further explore this attractive strategy for polypeptide synthesis, with our focus on two popular NCA monomers: BLG NCA and Boc-D,L-Lys NCA. LiHMDS

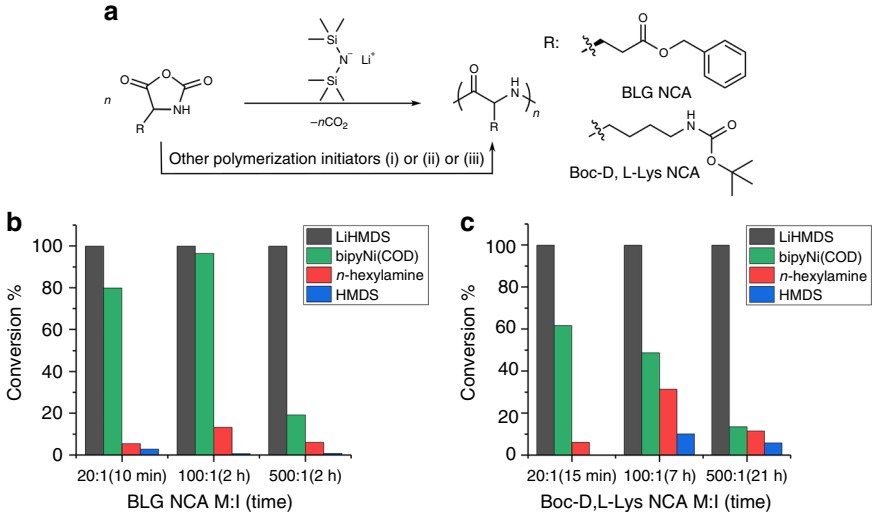

**Fig. 1** Ring-opening polymerization of NCA initiated by LiHMDS and other initiators. **a** NCA polymerization initiated by LiHMDS or other initiators in THF, initiator (i) n-hexylamine, initiator (ii) HMDS, initiator (iii) bipyNi(COD); **b** conversion of BLG NCA in LiHMDS, n-hexylamine, HMDS or bipyNi(COD) initiated polymerization at variable NCA:initiator ratios using THF as the reaction solvent; **c** conversion of Boc-D,L-Lys NCA in LiHMDS, n-hexylamine, HMDS, or bipyNi(COD) initiated polymerization at variable NCA:initiator ratios using THF as the reaction solvent

**Table 1 Characterization of poly-BLG prepared using different initiators**

| Entry | Initiator | M:I | Mn (g mol$^{-1}$) | Đ | DP |
|---|---|---|---|---|---|
| 1 | LiHMDS | 5 | 7490 | 1.23 | 34 |
| 2 | LiHMDS | 10 | 16,260 | 1.25 | 74 |
| 3 | LiHMDS | 20 | 27,340 | 1.20 | 124 |
| 4 | LiHMDS | 100 | 78,090 | 1.24 | 357 |
| 5 | LiHMDS | 500 | 111,900 | 1.15 | 510 |
| 6[a] | n-hexylamine | 20 | 4580 | 1.04 | 21 |
| 7 | n-hexylamine | 100 | 27,970 | 1.23 | 127 |
| 8[b] | n-hexylamine | 500 | 60,770 | 1.26 | 277 |
| 9 | HMDS | 20 | 7470 | 1.01 | 34 |
| 10 | HMDS | 100 | 32,070 | 1.22 | 146 |
| 11[b] | HMDS | 500 | 56,120 | 1.36 | 256 |
| 12 | bipyNi(COD) | 20 | 23,800 | 1.28 | 108 |
| 13 | bipyNi(COD) | 100 | 123,100 | 1.29 | 562 |
| 14 | bipyNi(COD) | 500 | 458,100 | 1.16 | 2091 |

*Note*: Polymerization of BLG NCA using LiHMDS as the initiator (entry 1–5) was compared to polymerization using n-hexylamine (entry 6–8), HMDS (entry 9–11), or bipyNi(COD) (entry 12–14) as the initiator. All polymers were synthesized using THF as the solvent and were characterized by GPC using DMF as the mobile phase. Mn is the obtained number average molecular weight, Đ is the dispersity index, DP is the obtained degree of polymerization
[a] a bimodal GPC trace was obtained with unexpected high Mn component (Fig. 2d for bimodal GPC trace)
[b] significant amount of NCA was left even after the polymerization reaction continued for 5 days

can initiate successful polymerization of BLG NCA to provided poly-BLG with narrow dispersities (Đ = 1.15–1.25) and variable chain lengths (DP = 34–510) (Table 1, entry 1–5; Fig. 2a for GPC traces). We also examined polymerization of BLG NCA using LiHMDS from different sources, different lots and different synthetic methods[63]. All six LiHMDS samples gave similar results in generating 350 mer poly-BLG (Supplementary Fig. 2, full scale GPC in Supplementary Figs. 90–95). Both n-hexylamine and HMDS were able to initiate controlled polymerization of BLG NCA and afford poly-BLG with short to moderate chain lengths (below 150-mer) and with narrow dispersities, although both reactions were slow, requiring 2–3 days for completion (Table 1, entry 6–7, 9–10). However, attempts to prepare 500-mer poly-BLG failed with either n-hexylamine or HMDS as initiator demonstrated that after 5 days, these reactions provided materials with average chain length <280-mer, and a large amount of NCA remained (Table 1, entry 8, 11). The transition metal catalyst bipyNi(COD) can initiate BLG NCA polymerization to prepare poly-BLG with medium to long chain lengths and narrow dispersities (Table 1, entry 12–14).

LiHMDS initiated the superfast polymerization of Boc-D,L-Lys NCA and provided polypeptides with variable average lengths, ranging from 19-mer to 405-mer, and narrow dispersities (Table 2, entry 1–5; Fig. 2b for GPC traces). N-hexylamine- and HMDS-initiated polymerizations of Boc-D,L-Lys NCA were much slower, paralleling the behavior noted above for BLG NCA. Yields and dispersities were good for shorter chain lengths; however, attempts to prepare 500-mer poly-Boc-D,L-Lys using n-hexylamine or HMDS were unsuccessful, as was the attempt with bipyNi(COD).

**Conformation analysis of poly-BLG**. To answer whether LiHMDS-initiated NCA polymerization can provide polypeptides with defined secondary structures, we studied the polymerization of BLG NCA as an example[64]. The solutions of poly-BLG at variable chain length (DP = 34, 74, 124) in hexafluoroisopropanol all displayed a positive band at 195 nm and two strong negative bands at 208 nm and 220 nm in circular dichroism (CD) spectra

(see details in Supplementary Methods), which indicates the secondary structure of α-helix (Fig. 2c). We also measured the specific rotations of deprotected poly-BLG (¹H NMR in Supplementary Fig. 24) by polarimetry to examine the possible backbone racemization during polymerization of BLG NCA[43]. We measured three batches of deprotected poly-BLG at pH 8.0 and obtained specific rotation values in the range of –78 to –83 deg·dm$^{-1}$·g$^{-1}$·mL that matches the reported specific rotation at pH 8.0 (around –80 deg·dm$^{-1}$·g$^{-1}$·mL) in literature[65]. An extra study indicated a linear relationship between the specific rotation (in a range of 0–80 deg·dm$^{-1}$·g$^{-1}$·mL) and the degree of backbone racemization (Supplementary Fig. 6). Therefore, backbone racemization is not a concern during LiHMDS-initiated NCA polymerization, though we can not totally rule out this possibility. This conclusion is also consistent with that drawn from CD analysis aforementioned. It is worth mentioning that use of hexylamine as the initiator to prepare 20-mer polymer from BLG NCA or Boc-D,L-Lys, led to materials that displayed puzzling bimodal GPC traces (Fig. 2d, e). Comparable observations have been reported by other groups in reactions initiated with primary amines[37,66,67]. Neither LiHMDS- nor HMDS-initiated polymerizations of BLG NCA and Boc-D,L-Lys NCA produced samples with bimodal GPC traces. The bimodal GPC data for n-hexylamine-initiated polymerization of NCA were likely not the effect of secondary structures of resulting polypeptides because we used the Boc-D,L-Lys NCA, a racemic mixture rather than a homochiral NCA. Polypeptides prepared from the racemic Boc-D,L-Lys NCA are unlikely to have secondary structures like α-helix and β-sheet.

**Kinetics analysis**. We studied the kinetics of LiHMDS-initiated polymerization for BLG NCA, Boc-D,L-Lys NCA and Boc-L-Lys NCA. The rate of polymerization, $k_p$[I], was calculated from a linear fitting of the plot that depicts natural logarithm of monomer concentration change vs. reaction time. BLG NCA polymerization initiated by LiHMDS, n-hexylamine or bipyNi (COD) in THF had $k_p$[I] values of 1.69 h$^{-1}$ (Fig. 2f), 0.07 h$^{-1}$ or 0.87 h$^{-1}$, respectively (Supplementary Fig. 5). LiHMDS-initiated polymerization of Boc-D,L-Lys NCA or Boc-L-Lys NCA in THF had $k_p$[I] values of 0.33 h$^{-1}$ or 1.57 h$^{-1}$, respectively (Fig. 2f). Thus, the homochiral starting materials, BLG NCA and Boc-L-Lys NCA, displayed comparable reactivity, but the racemic NCA reacted more slowly.

**LiHMDS-initiated open vessel NCA polymerization**. Continuous studies demonstrated that LiHMDS-initiated open vessel NCA polymerization is compatible with common NCA monomers such as Boc-L-Lys NCA, Nδ-tert-butyloxycarbonyl-L-ornithine (Boc-L-Orn) NCA, O-(tert-butyl)-L-serine (*t*Bu-L-Ser) NCA, L-Leucine (L-Leu) NCA, and β-tert-butyl-L-aspartate (*t*Bu-L-Asp) NCA. Homo-polymerization of these NCAs provided poly-BLG, poly-Boc-L-Lys, and poly-Boc-L-Orn with narrow dispersities (Đ = 1.08–1.28) and variable chain lengths (Table 3, entry 1–11; Fig. 3b–d for GPC traces). The highest number average molecular weight (Mn) obtained from GPC characterization on poly-BLG and poly-Boc-L-Lys was 283,400 g mol$^{-1}$ and 111,000 g mol$^{-1}$, respectively. We also demonstrated successful NCA co-polymerization using LiHMDS as the initiator in open vessels. Polymerization on 1:1 mixture of two NCA monomers, (Boc-L-Lys NCA + *t*Bu-L-Ser NCA), (Boc-L-Orn NCA + L-Leu NCA) and (BLG NCA + *t*Bu-L-Asp NCA), provided corresponding random co-polymers with narrow dispersities (Table 3, entry 12–14, Fig. 3e for GPC traces). Moreover, LiHMDS-initiated open vessel NCA polymerization is capable to prepare block co-polymers as demonstrated using Boc-L-Lys

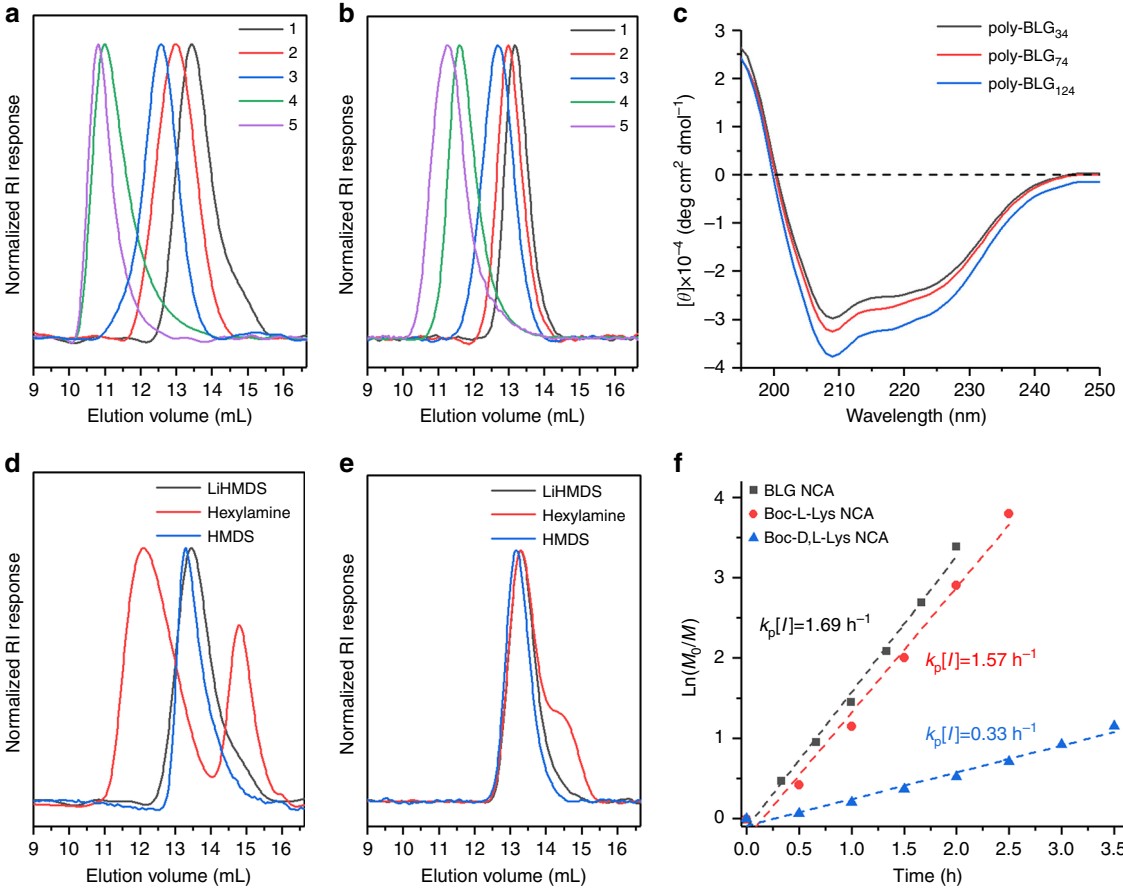

**Fig. 2** Characterization of polypeptides synthesized from LiHMDS-initiated NCA polymerization. **a** GPC traces of poly-BLG corresponding to entry 1–5 in Table 1; **b** GPC traces of poly-Boc-D,L-Lys corresponding to entry 1–5 in Table 2; **c** CD spectra of poly-BLG at variable DP prepared from LiHMDS-initiated NCA polymerization; **d** single peak GPC traces and bimodal GPC trace obtained from LiHMDS, HMDS and n-hexylamine initiated BLG NCA polymerization, respectively; **e** single peak GPC traces and bimodal GPC trace obtained from LiHMDS, HMDS and n-hexylamine initiated Boc-D,L-Lys NCA polymerization, respectively; **f** reaction rates of LiHMDS initiated BLG NCA, Boc-L-Lys NCA and Boc-D,L-Lys NCA polymerization in THF with NCA:initiator ratio of 100:1 and initial NCA concentration at 0.2 M; $k_p[I]$ represents the rate of polymerization reaction, whereas $k_p$ is the rate constant of the polymerization and $[I]$ is the concentration of initiator. All full scale GPC traces were included in the Supplementary Figs. 45–72

### Table 2 Characterization of poly-Boc-D,L-Lys prepared using different initiators

| Entry | Initiator | M:I | Mn (g mol$^{-1}$) | Đ | DP |
|---|---|---|---|---|---|
| 1 | LiHMDS | 5 | 4230 | 1.19 | 19 |
| 2 | LiHMDS | 10 | 7250 | 1.16 | 32 |
| 3 | LiHMDS | 20 | 12,370 | 1.23 | 54 |
| 4 | LiHMDS | 100 | 36,300 | 1.23 | 159 |
| 5 | LiHMDS | 500 | 92,340 | 1.26 | 405 |
| 6[a] | n-hexylamine | 20 | 5980 | 1.35 | 26 |
| 7 | n-hexylamine | 100 | 15,800 | 1.23 | 69 |
| 8[b] | n-hexylamine | 500 | 17,030 | 1.14 | 75 |
| 9 | HMDS | 20 | 4250 | 1.10 | 18 |
| 10 | HMDS | 100 | 16,630 | 1.25 | 73 |
| 11[b] | HMDS | 500 | 18,140 | 1.10 | 80 |
| 12 | bipyNi(COD) | 20 | 14,600 | 1.25 | 64 |
| 13 | bipyNi(COD) | 100 | 17,170 | 1.20 | 75 |
| 14[b] | bipyNi(COD) | 500 | 5790 | 1.20 | 25 |

*Note*: Polymerization of Boc-D,L-Lys NCA using LiHMDS as the initiator (entry 1–5) was compared to polymerization using n-hexylamine (entry 6–8), HMDS (entry 9–11) or bipyNi (COD) (entry 12–14) as the initiator. All polymers were synthesized using THF as the solvent and were characterized by GPC. Mn is the obtained number average molecular weight, Đ is the dispersity index, DP is the obtained degree of polymerization
[a] a bimodal GPC trace was obtained with unexpected low Mn component (Fig. 2e for bimodal GPC trace)
[b] significant amount of NCA was left even after the polymerization reaction continued for 5 days

### Table 3 Polypeptides synthesized from LiHMDS-initiated open vessel NCA polymerization

| Entry | Polymer | M:I | Mn (g mol$^{-1}$) | Đ | DP |
|---|---|---|---|---|---|
| 1 | poly-BLG | 5 | 6440 | 1.23 | 30 |
| 2 | poly-BLG | 20 | 31,470 | 1.28 | 143 |
| 3 | poly-BLG | 100 | 88,630 | 1.28 | 404 |
| 4 | poly-BLG | 500 | 153,100 | 1.11 | 699 |
| 5 | poly-BLG | 1000 | 283,400 | 1.08 | 1294 |
| 6 | poly-Boc-L-Lys | 5 | 6900 | 1.20 | 30 |
| 7 | poly-Boc-L-Lys | 20 | 13,550 | 1.24 | 59 |
| 8 | poly-Boc-L-Lys | 100 | 62,780 | 1.25 | 275 |
| 9 | poly-Boc-L-Lys | 500 | 111,000 | 1.18 | 486 |
| 10 | poly-Boc-L-Orn | 5 | 6090 | 1.23 | 28 |
| 11 | poly-Boc-L-Orn | 20 | 11,780 | 1.24 | 55 |
| 12 | poly(Boc-L-Lys)-r-poly (tBu-L-Ser) | 5 | 7200 | 1.24 | 41 |
| 13 | poly(Boc-L-Orn)-r-poly(L-Leu) | 5 | 6420 | 1.24 | 39 |
| 14 | poly(BLG)-r-poly(tBu-L-Asp) | 5 | 6020 | 1.25 | 35 |
| 15 | poly-Boc-L-Lys (1st block) | 5 | 6160 | 1.23 | 27 |
|   | poly(Boc-L-Lys)-b-poly(Boc-L-Lys) |   | 13,980 | 1.29 | 61 |

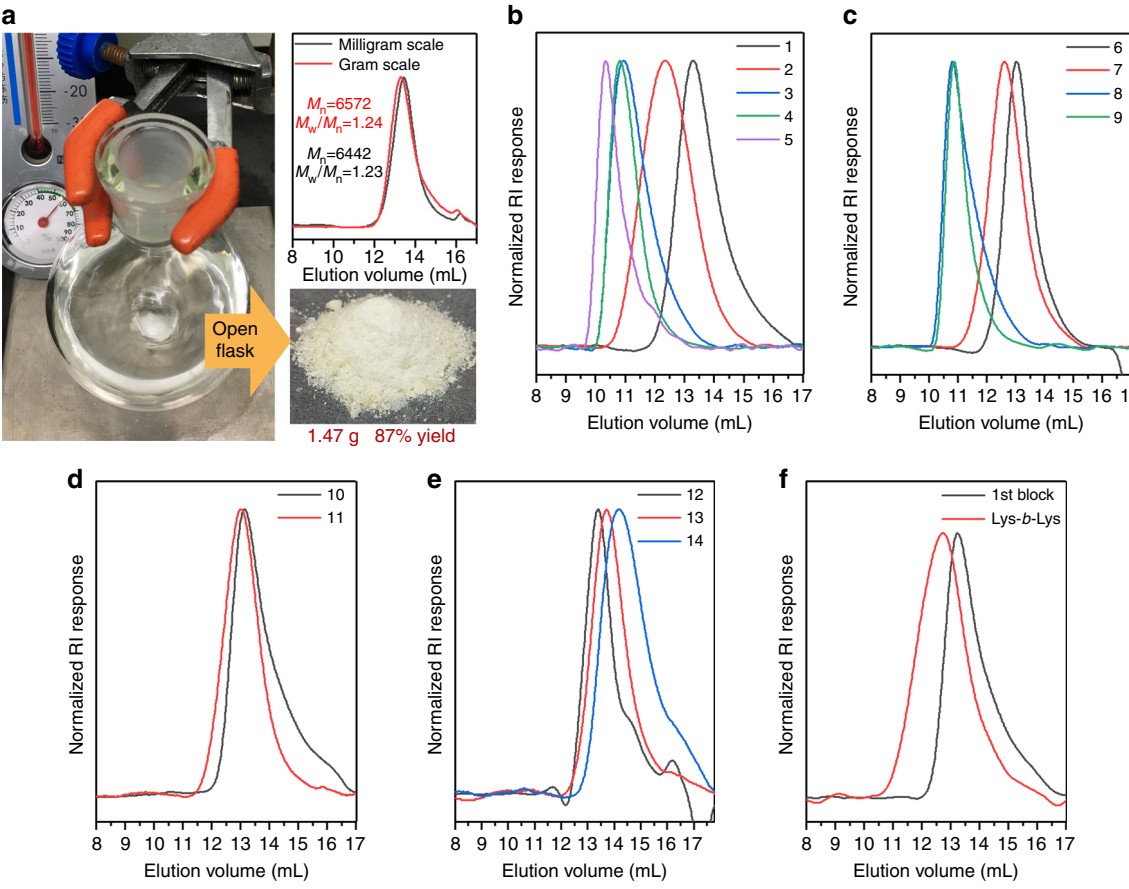

**Fig. 3** LiHMDS-initiated open vessel polymerization of various NCAs. **a** LiHMDS-initiated open vessel polymerization of BLG NCA and corresponding GPC traces of resulting poly-BLG in THF at 26 mg and 2 g scale, respectively; **b** GPC traces of polypeptides polymerized from BLG NCA corresponding to entry 1–5 in Table 3; **c** GPC traces of polypeptides polymerized from Boc-L-Lys NCA corresponding to entry 6–9 in Table 3; **d** GPC traces of polypeptides polymerized from Boc-L-Orn NCA corresponding to entry 10–11 in Table 3; **e** GPC traces of random co-polymers synthesized from a 1:1 mixture of two NCA monomers corresponding to entry 12–14 in Table 3; **f** GPC traces of the block copolymer corresponding to entry 15 in Table 3. All full scale GPC traces were included in the Supplementary Figs. 75–89

NCA. After a quick completion for the first block (DP = 27, Đ = 1.23), adding extra Boc-L-Lys NCA to the reaction resulted in successful extension of the polymer to give the final block co-polymer (DP = 61 and Đ = 1.29) (Table 3, entry 15; Fig. 3f for GPC traces).

Conventional NCA polymerizations are hindered by moisture; therefore, these reactions are generally conducted using a Schlenk line or a glovebox to maintain a dry environment. We were surprised to discover that LiHMDS-initiated BLG NCA poly-merization could be performed in an open flask, in an environment with 60% relative humidity. Polymerizations at milligram scale or gram scale were complete within 5 min and afforded 30-mer poly-BLG, in 81% or 87% yield, respectively, with narrow dispersities (Đ = 1.23–1.24) as described in Fig. 3a. These findings are important because it is vastly more convenient to conduct NCA polymerizations if a glove box or Schleck line is unnecessary.

**Mechanism study on LiHMDS-intiated NCA polymerization.** LiHMDS is a non-nucleophilic base, which implies an anionic ring opening polymerization mechanism of LiHMDS-initiated NCA polymerization (Fig. 4a). In the initiation step, LiHMDS may deprotonate either the 3-NH or the 4-CH in a NCA ring. To figure out if the initiation follows one or both pathways, we did FT-IR characterization on a 1:1 mixture of LiHMDS and Boc-D, L-Lys NCA in THF (Fig. 4b, d). In the FT-IR spectrum of Boc-D,

L-Lys NCA, peaks at 1786 $cm^{-1}$ and 1875 $cm^{-1}$ represent two carbonyl groups in the NCA ring. In the IR spectrum of 1:1 Boc-D,L-Lys NCA/LiHMDS mixture, the disappearance of these two peaks indicates the ring opening of NCA; the emergence of peaks at 2231 $cm^{-1}$ and 1599 $cm^{-1}$ confirms the formation of iso-cyanate and carboxylate groups, respectively, in the intermediate **2** (Fig. 4b, d). Above observation, in addition to the lack of IR signal for ketene from 4-CH deprotonation pathway indicates that LiHMDS initiates NCA polymerization by deprotonating the 3-NH rather than the 4-CH in the NCA ring. Further HRESI-MS analysis on the 1:1 Boc-D,L-Lys NCA/LiHMDS mixture found fragments at $m/z$ 279.1532 and 253.1729, indicating the trans-formation of NH-deprotonated Boc-D,L-Lys NCA **1** to isocyanate intermediate **2** and a following conversion of the intermediate **2** to free amine fragment **3** (Fig. 4e). In addition, the observation of fragments at $m/z$ 551.3044 and 525.3112 indicates the formation of dimer **4** and the following conversion of **4** to fragment **5**. During polymerization, once intermediate 4 was formed, the N-terminal lithium carbamate can undergo nucleophile addition to the C5-carbonyl of a new NCA for chain propagation (Fig. 4c). This proposed mechanism for chain propagation was supported by the observation of tripeptide **6**, having a characteristic N-terminal lithium carbamate group at $m/z$ 779.4641 in HRESI-MS (Fig. 4c, e). We hypothesized the N-terminal lithium carbamate in intermediate **4**, as a weak base, has a low potency to deprotonate 3-NH in a NCA ring and generate an activated NCA that will

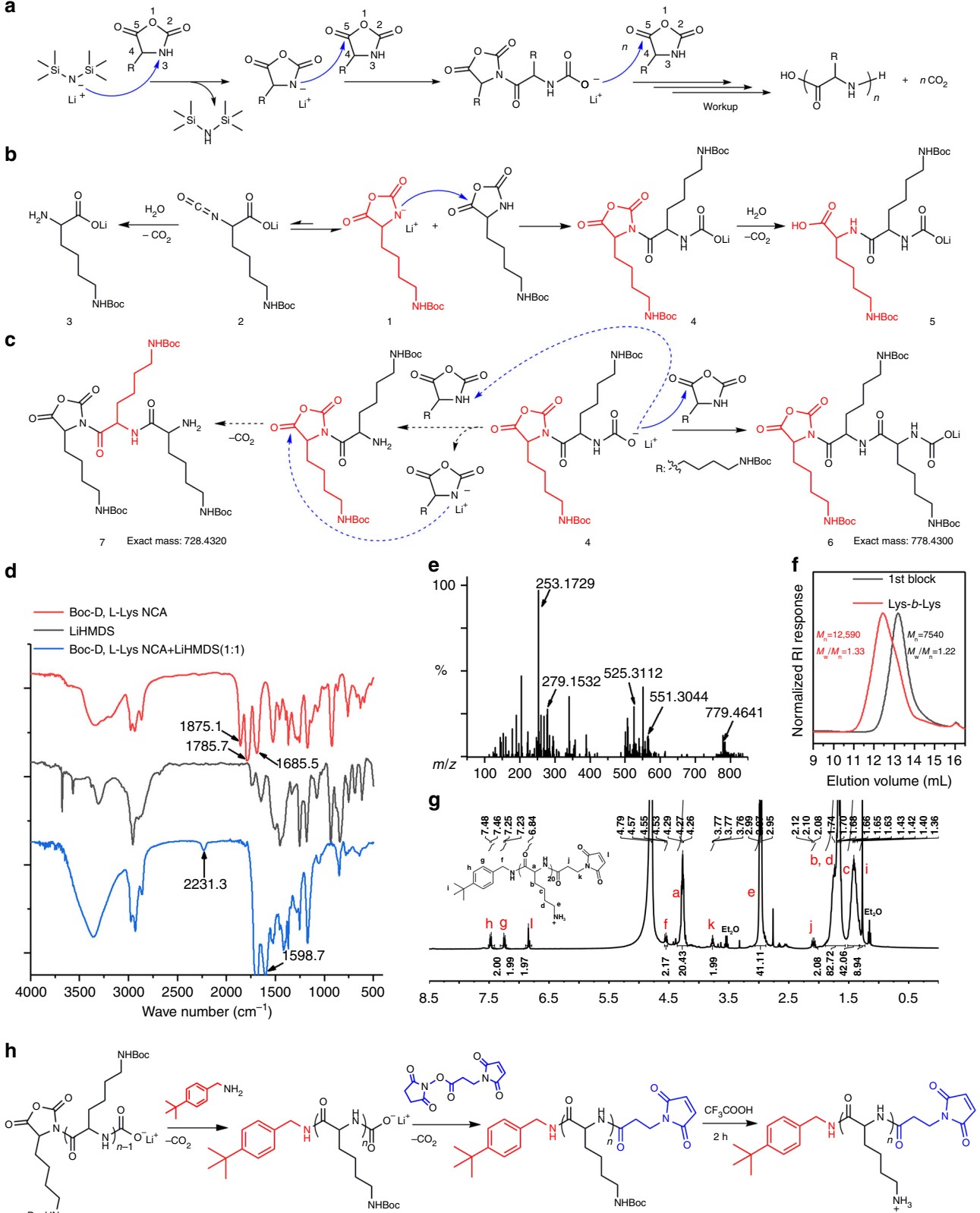

**Fig. 4** Mechanism study of LiHMDS-initiated NCA polymerization. **a** Proposed mechanism of LiHMDS-initiated NCA polymerization; **b** transformation of Boc-D,L-Lys NCA in a 1:1 Boc-D,L-Lys NCA/LiHMDS mixture; **c** comparison between two possible chain propagation pathways; **d** FT-IR spectra of Boc-D,L-Lys NCA, LiHMDS and a 1:1 Boc-D,L-Lys NCA /LiHMDS mixture; **e** HRESI-MS analysis on a 1:1 Boc-D,L-Lys NCA/LiHMDS mixture; **f** GPC characterization on block co-polymer of poly(Boc-D,L-Lys)-b-poly(Boc-D,L-Lys) (Lys-b-Lys) obtained from LiHMDS-initiated polymerization of Boc-D,L-Lys NCA. Full scale GPC traces were included in the Supplementary Fig. 73-74. **g** Proton NMR characterization on terminal functionalization of a 20-mer polypeptide (details see Supplementary Fig. 37) 11; **h** reaction scheme for N-terminal and C-terminal double functionalization of a 20-mer polypeptide to give polymer **11**, where $n = 20$ in the reaction scheme

result in tripeptide **7** via nucleophilic addition and decarboxylation steps (Fig. 4c). As expected, no signal of tripeptides **7** was found in HRESI-MS spectrum (Fig. 4e). In addition, MALDI-TOF-MS characterization on purified poly-BLG confirmed the presence of a characteristic N-terminal lithium carbamate group (Supplementary Fig. 7a). These data all support a chain propagation mechanism with N-terminal lithium carbamate as the reactive center[68]. Both FT-IR and HRESI-MS data support an anionic ring-opening polymerization mechanism, where LiHMDS initiates the NCA polymerization by deprotonating 3-NH in the NCA ring for the first step, followed by further nucleophilic addition of N-terminal carboxylate to C5-carbonyl of another NCA (Fig. 4a). A block copolymer, poly(Boc-D,L-Lys)-*b*-poly(Boc-D,L-Lys), was successfully synthesized by adding the Boc-D,L-Lys NCA into the reaction in two portions sequentially (Fig. 4f). This result indicates a living polymerization property of LiHMDS-initiated NCA polymerization.

Proton NMR characterization on purified poly-BLG obtained a 1:5 ratio for protons in backbone methylene group vs. sidechain benzyl group (Supplementary Fig. 7b). This observation indicates that once the anionic NCA was generated by LiHMDS during the initiation step, it selectively attacked another NCA monomer rather than the side chain benzyl ester because the anhydride carbonyl group is much more reactive than benzyl ester carbonyl group for nucleophilic addition. As the reactive center during polymer chain propagation, the N-terminal carbamate may possibly attacked the C-terminal NCA moiety and form cyclic polypeptides. The resulting cyclic polypeptides were known to be found easily in MALDI-TOF-MS analysis[60]. However, no signal of cyclic polypeptide was found in the MALDI-TOF-MS spectrum of obtained poly-BLG from LiHMDS-initiated polymerization (Supplementary Fig. 7a). We also found no observable cyclization at the end extremity as a possible side reaction during propagation of LiHMDS-initiated BLG NCA polymerization (Supplementary Fig. 38).

According to the proposed mechanism of LiHMDS-initiated NCA polymerization, the resulting polypeptides have a C-terminal NCA group and an N-terminal lithium carbamate group that enable easy functionalization at both ends. We examined this hypothesis using LiHMDS-initiated polymerization of Boc-D,L-Lys NCA as a demonstration to prepared end functionalized poly-Boc-D,L-Lys at 20-mer. To tune the C-terminal end group we added 4-tert-butylbenzylamine, a nucleophile, to the completed polymerization mixture and obtained poly-Boc-D,L-Lys with the expected C-terminal 4-tert-butylbenzyl group (Fig. 4h, polymer **9**). To further tune the N-terminal end group we mixed N-succinimidyl 3-maleimidopropionate, an electrophile, with above polymer and obtained poly-Boc-D,L-Lys with an N-terminal maleimide group (Fig. 4h, polymer). Proton NMR characterization obtained an integration ratio of 2:2:20 for $H_h$(C-terminal Benzyl):$H_l$(N-terminal maleimide):$H_a$(backbone CH), which indicates a successful and controlled terminal functionalization on both ends of this 20-mer polypeptide to have almost one C-terminal 4-tert-butylbenzyl group and one N-terminal maleimide group per polypeptide chain (Fig. 4g). This result also supports aforementioned statement that acyclic polymers dominate in LiHMDS-initiated NCA polymerization, though we cannot totally rule out the possibility in forming cyclic polypeptides.

**Demonstration of LiHMDS-initiated NCA polymerization in polypeptide library synthesis**. The LiHMDS-initiated fast NCA polymerization is an excellent fit for rapid synthesis of polypeptide libraries for high-throughput functional screening. We did a proof-of-concept demonstration on quick library synthesis of hydrophobic/cationic amphiphilic co-polymers as mimics of host defense peptide

(HDP) for antibacterial activity screening (Fig. 5). To address the challenge of low stability upon proteolysis, one of the innate and prominent shortcomings of HDP, we used racemic Boc-D,L-Lys NCA rather than homochiral Boc-L-Lys NCA as the cationic subunit within the antimicrobial polypeptides. These amphiphilic co-polymers were quickly synthesized from LiHMDS-initiated polymerization using a mixture of BLG NCA and Boc-D,L-Lys NCA. The resulting polypeptides, at about 26-mer, have low dispersities (Đ = 1.16–1.17) and incrementally increased ratios of the cationic subunit (Fig. 5a, b) (NMR and full GPC traces are shown in Supplementary Figs. 25–36). The large difference in reaction rate between BLG NCA and racemic Boc-D,L-Lys NCA (Fig. 2f) urged us to explore NCA subunit composition of resulting polypeptides. Analyzing on subunit composition within polymer chains along the polymerization progress (or along the NCA conversion) indicated a higher content of BLG than Boc-D,L-Lys at the C-terminus before 30% consumption of the total NCA monomers. As a consequence, more Boc-D,L-Lys gets into the polymer chain than does BLG at the N-terminal after 80% consumption of total NCA monomers (Fig. 5c) (see details in Supplementary Methods). It is noteworthy that almost equal ratio of two subunits exists for major part of the polymer in middle of the polymer chain. Although the resulting polymers are not globally random co-polymer, they are still valid as antimicrobial mimics of HDP because our previous study confirmed that some extent of intermixing of two subunits is enough to mimic HDP[54]. This result also implies that random co-polymers can be prepared in LiHMDS-initiated NCA polymerization if NCA monomers with close reaction rate are used for a combination. The obtained co-polymers from the mixture of BLG NCA and Boc-D,L-Lys NCA were subjected to deprotection and the final polypeptides were evaluated for their activities against multiple bacteria, including *Acinetobacter baumannii* (*A. baumannii*), *Escherichia coli* (*E. coli*), *Bacillus subtilis*, (*B. subtilis*), and multi-drug resistant *P. aeruginosa* and MRSA. A plot of these polypeptides' antibacterial activities, the minimum inhibitory concentrations (MICs), against ratios of subunits within polymers indicates that amphiphilic polypeptides can have strong and broad-spectrum antimicrobial activities if having an appropriate subunit ratio, 30–50 molar percentage of the hydrophobic subunit in this study (Fig. 5d). This proof-of-concept demonstration implies that the superfast NCA polymerization developed in this study enables quick library synthesis of polypeptides for high through-put functional screening to find potent and broad-spectrum antimicrobial polypeptides in therapeutic and biomedical applications.

## Discussion
In this study, we report a LiHMDS-initiated superfast NCA polymerization that can afford polypeptides with variable lengths (20–1294 mers) and narrow distributions of molecular weight. Conventional NCA polymerization strategies are highly sensitive to moisture and are generally conducted using a Schlenk line or a glovebox to maintain a dry environment. In sharp contrast, LiHMDS-initiated NCA polymerization can be operated in an open flask outside of the glovebox without any protection. Successful open vessel polymerizations can be achieved under environmental conditions at least in 60% relative humidity using untreated THF. This method also enables quick library synthesis and large scale synthesis of polypeptides. Moreover, LiHMDS-initiated polymerization can be used for convenient synthesis of long polypeptides. Owing to this NCA polymerization strategy, we quickly synthesized a library of polypeptide and identified polypeptides with broad-spectrum and potent activities against bacteria, including multi-drug resistant *S. aureus* and *P. aeruginosa*. This demonstration validates LiHMDS-initiated NCA polymerization for its capability and potential in combinational chemistry of NCAs to boost the exploration and application of polypeptides in diverse fields.

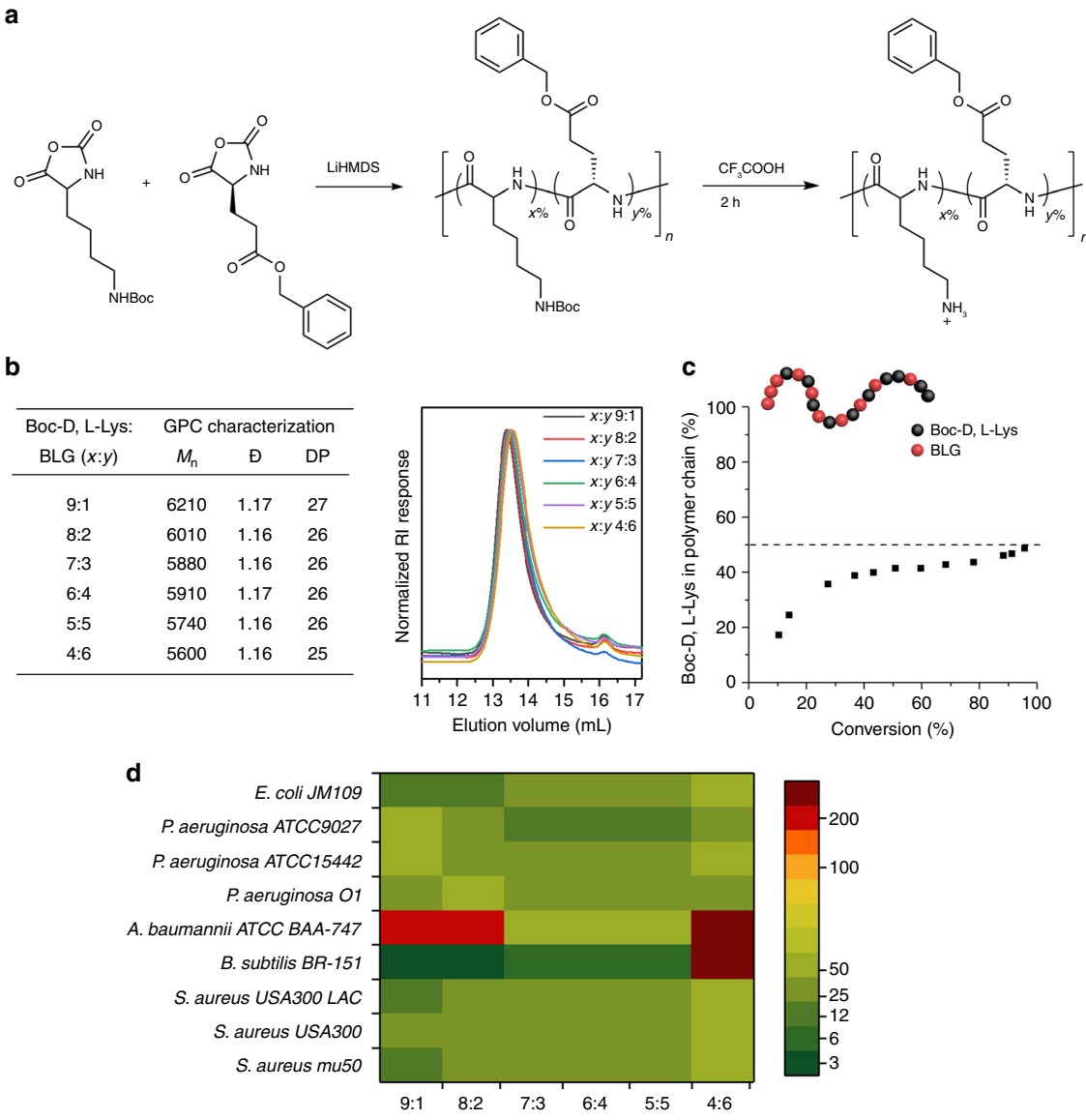

**Fig. 5** Fast library synthesis of polypeptides for antibacterial activity studies. **a** LiHMDS-initiated polymerization of a mixture of BLG NCA and Boc-D,L-Lys NCA in variable NCA ratios, **b** characterization and GPC traces of resulting polymers at the NHBoc protected stage. Mn is the obtained number average molecular weight, Đ is the dispersity index, DP is the obtained degree of polymerization; **c** plot of subunit composition within polymer chains against total NCA conversion; **d** plot of deprotected polypeptides' antimicrobial activities (MIC) against ratios of subunits (x:y Boc-D,L-Lys:BLG) within these co-polymers, with BLG, the hydrophobic subunit, varied from 10% to 60%

## Methods

**Measurements**. $^{1}$H NMR and $^{13}$C NMR spectra were collected on a Bruker spectrometer at 400 MHz and 100 MHz, respectively. $^{1}$H NMR chemical shifts were referenced to the resonance for residual protonated solvent ($\delta$ 7.26 for CDCl$_3$, $\delta$ 4.79 for D$_2$O, $\delta$ 2.50 for DMSO-$d6$, $\delta$ 1.94 for CD$_3$CN). $^{13}$C NMR chemical shifts were referenced to the resonance for residual solvent ($\delta$ 77.16 for CDCl$_3$, $\delta$ 1.32 for CD$_3$CN). Gel permeation chromatography (GPC) was performed on a Waters GPC instrument equipped with a Waters 1515 isocratic HPLC pump, a Brookhaven BI-MwA multi-angle light scattering detector and a Waters 2414 refractive index detector using DMF supplemented with 0.01 M LiBr as the mobile phase at a flow rate of 1 mL/min at 50 °C. The GPC were equipped by a Tosoh TSKgel Alpha-2500 column (particle size 7 μm) and a Tosoh TSKgel Alpha-3000 column (particle size 7 μm) linked in series. Relative number-average molecular weight (Mn) and dispersity index (Đ) were calculated from a calibration curve using PMMA as standards. Absolute Mn and Đ were calculated using BIC ParSEC Software with a dn/dc value of 0.1 mL/g for all polymers. Fourier transform infrared (FT-IR) spectra were recorded on a Thermo Electron Nicolet 6700 FT-IR spectrophotometer by using a KBr plate. High resolution ESI-MS (HRESI-MS) was collected on a Waters XEVO G2 TOF mass spectrometer. High performance liquid chromatography (HPLC) analysis was performed with SHIMADZU LC 20AR HPLC System equipped with a

Gemini 5 μm NX-C$_{18}$ column. HPLC was used to monitor NCA stability in different solvents and polymerization progress. To examine if the water in the HPLC eluent have significant effect on the result of analysis during the short time of HPLC elution, the stability studies on NCA in 90% and 60% acetonitrile (Supplementary Fig. 3) and the calibration curves (Supplementary Fig. 4) were conducted. Matrix-assisted laser desorption ionization time of flight (MALDI-TOF) mass spectra were collected using an AB SCIEX 4800plus MALDI-TOF analyzer in linear mode equipped with a nitrogen laser emitting at 337 nm. 2,5-Dihydroxybenzoic acid (DHB) was used as matrices. Circular Dichroism (CD) was performed with a J-1500 CD Spectrometer (JASCO Corporation) using a temperature-controlled cuvette holder (Quantum Northwest, model TC 125). Chirality of the deprotected polymers at $\lambda = 589$ nm was recorded using the Autopol® V automatic polarimeter at pH 8 and 25 °C. Water content was measured by a Mettler Toledo V20 volumetric Karl-Fischer titrator. Inductively coupled plasma-mass spectrometry (ICP-MS) samples were run on a PerkinElmer NexION® 2000 instrument.

**Open vessel NCA polymerization initiated by LiHMDS**. To a solution of BLG NCA (26.3 mg, 0.1 mmol) in THF in a flask open to air at about 60% relative humidity outside of a glovebox, was added a 0.1 M solution of LiHMDS in THF (0.2 mL) at room temperature under stirring. The reaction mixture was stirred at

room temperature for 5 min and then the resulting polypeptide was precipitated out by pouring the reaction mixture into cold petroleum ether. The precipitation was collected after centrifugation to remove the solvent and dried under vacuum to afford poly-BLG as a white solid 17.7 mg in 81% yield. Gram scale (2.01 g, 7.63 mmol) open vessel BLG NCA polymerization was operated by following above operation with reaction stirred at room temperature for 10 min. The purified poly-BLG was obtained 1.47 g in 87% yield. The operation for open vessel synthesis of poly-BLG in variable chain length was controlled via the BLG NCA:initiator ratio. BLG NCA was weighed out and dissolved in THF to final concentration at 0.1 M to 0.5 M, depending on target polypeptide chain length, in an open flask equipped with a magnetic stir bar outside of a glovebox. Then a solution of LiHMDS in THF (0.02 M or 0.1 M) was added to the reaction quickly. The reaction mixture was stirred at room temperature for 5 mins to 2 h depending on the targeting poly-peptide chain length. Poly-$BLG_{30}$ was synthesized from a mixture of BLG NCA (26.3 mg, 0.1 mmol) and a 0.1 M solution of LiHMDS in THF (0.2 mL) to reach a final BLG NCA concentration at 0.1 M. Poly-$BLG_{143}$ was synthesized from a mixture of BLG NCA (26.3 mg, 0.1 mmol) and a 0.1 M solution of LiHMDS in THF (0.05 mL) to reach a final BLG NCA concentration at 0.1 M. Poly-$BLG_{404}$ was synthesized from a mixture of BLG NCA (52.6 mg, 0.2 mmol) and a 0.02 M solution of LiHMDS in THF (0.1 mL) to reach a final BLG NCA concentration at 0.5 M. Poly-$BLG_{600}$ was synthesized from a mixture of BLG NCA (52.6 mg, 0.2 mmol) and a 0.02 M solution of LiHMDS in THF (0.02 mL) to reach a final BLG NCA concentration at 0.5 M. The poly-$BLG_{1294}$ was synthesized from a mixture of BLG NCA (105.2 mg, 0.4 mmol) and a 0.02 M solution of LiHMDS in THF (0.02 mL) to reach a final BLG NCA concentration at 1 M. Homopolymerization of other NCA monomers (Boc-L-Lys NCA and Boc-L-Orn NCA) were also operated in open vessels using LiHMDS as the initiator by following the protocol of BLG NCA described above. Open vessel synthesis of random co-polymers was performed using the mixture of two NCA monomers (1:1 Boc-L-Lys NCA:tBu-L-Ser NCA, 1:1 Boc-L-Orn NCA:L-Leu NCA, and 1:1 BLG NCA:tBu-L-Asp NCA). Taking one random co-polymerization for example, Boc-L-Lys NCA (27.2 mg, 0.1 mmol) and tBu-L-Ser NCA (18.7 mg, 0.1 mmol) were weighed out and dissolved in THF (2 mL) in a flask equipped with a magnetic stir bar and open to air. Then a 0.1 M solution of LiHMDS in THF (0.4 mL) was added to the reaction. The reaction mixture was stirred at room temperature for 5 min to get complete conversion of the monomers. After diluting the solution to 5 mg/mL using DMF (containing 0.01 M LiBr), the obtained solution was analyzed by GPC to measure the absolute molecular weight (Supplementary Fig. 40). Then the polymer was deprotected to measure the composition using NMR (Supplementary Fig. 39). Random co-polymer of poly(Boc-L-Orn)$_{0.5}$-r-poly(L-Leu)$_{0.5}$ was synthesized in an open vessel using the mixture of Boc-L-Orn NCA (25.8 mg, 0.1 mmol) and L-Leu NCA (15.7 mg, 0.1 mmol). Random co-polymer of poly(BLG)$_{0.5}$-r-poly(tBu L-Asp)$_{0.5}$ was synthesized in an open vessel using the mixture of BLG NCA (26.3 mg, 0.1 mmol) and tBu-L-Asp NCA (21.5 mg, 0.1 mmol). The open vessel synthesis and characterization of both random co-polymers (Supplementary Figs. 41-44) was performed by following the protocol similar to that of poly(Boc-L-Lys)$_{0.5}$-r-poly(tBu-L-Ser)$_{0.5}$ described above. For the open vessel synthesis of block co-polymer, Boc-L-Lys NCA (27.2 mg, 0.1 mmol) was dissolved in THF (0.7 mL) followed by addition of 0.1 M solution of LiHMDS in THF (0.2 mL). The reaction was stirred in a flask open to air at room temperature for 2 min to get the first block. Then Boc-L-Lys NCA (27.2 mg, 0.1 mmol) in THF (0.2 mL) was added to the reaction mixture and the reaction was continued for 2 more hours to give the block polymer poly(Boc-L-Lys)-b-poly(Boc-L-Lys). These polypeptides were characterized by GPC using DMF as the mobile phase.

## Data availability

Data that support the findings detailed in this study are available in the supple-mentary information and this article. Any other source data perceived as pertinent are available, on reasonable request, from the corresponding author.

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

## Acknowledgements

This research was supported by the National Natural Science Foundation of China for Innovative Research Groups (No. 51621002), the National Natural Science Foundation of China (No. 21574038, 21774031), the National Key Research and Development Program of China (2016YFC1100401), the "Eastern Scholar Professorship" from Shanghai local government (TP2014034), the 1000 talent young scholar program in China, 111 project (B14018), the national special fund for State Key Laboratory of Bioreactor Engineering (2060204), the Fundamental Research Funds for the Central Universities (22221818014), the program for professor of special appointment at ECUST. The authors also thank Research Center of Analysis and Test of East China University of Science and Technology for the help on the characterization. The authors thank Professor Samuel H. Gellman for invaluable discussions on the manuscript.

## Author contributions

Y.W. and R.L. conceived the idea, proposed the strategy, designed the experiments, evaluated the data, and wrote the manuscript together. Y.W. performed majority of the experiments. D.Z. participated NCA synthesis and initial polymerization test. P.M. conducted the antimicrobial assay. R.Z. participated HPLC analysis. L.H. participated MassSpec spectra analysis. All authors proof read the manuscript.

## Additional information

**Competing interests:** R.L. and Y.W. are co-inventors on a patent covering reported NCA ring opening polymerization. All remaining authors declare no competing interests.

