## [Peer Review File · Nature Communications]

Reviewers' comments:

Reviewer #1 (Remarks to the Author):

In this manuscript, Dr Liu and his co-workers reported an easy handled and superfast ring opening polymerization of NCA by using LiHMDS as initiator. This is a quite interesting research direction, considering the wide application of the peptidomimics as functional biomaterials. The authors claim they developed a LiHMDS initiated superfast NCA polymerization that can finish within a range of mins to hours and this polymerization method is tolerant to moisture, so that it can be operated successfully in open vessels outside of a glovebox without any protection. They have synthesized up to 500 DP of polypeptides with low polydispersity (PDI=1.15-1.26) and they also demonstrated a successful open-flask polymerization using this method, obtaining 1.47g polymers. Though the results are promising, the fast polymerization of NCA has been illustrated quite long time ago using the organonickel compounds (NATURE | VOL 390 | 27 NOVEMBER 1997). The rate constant is $2.7 \times 10^{-4} \text{ s}^{-1}$ (0.67 mM initiator) compared to 1.7 h^{-1} (2 mM initiator). The polymerization speed of these two methods is at least comparable. But considering the toxicity issue of transition metal used, the finding of LiHMDS as initiator is still quite interesting, especial for preparing functional biomaterials. The most important finding is the high tolerant polymerization method and it can be carried outside of glovebox. It will be much easier to carry out the NCA polymerization by this method. It will be a useful new NCA polymerization technique and will be interesting to a broad field, from polymer chemistry, material and biological science.

However, there are some limitations that the authors should be addressed before its further consideration for publication.

1. The open-flask NCA polymerization is very impressive, but it was only conducted to synthesize a low molecular weight of PBLG. It is highly advised that the author can conduct a more complete study of this moisture tolerant NCA polymerization method. For example, what is the highest DP of polymers that can be prepared via this method? Can the polymer chain still be further extended when conducted outside the glovebox? Does this polymerization method is monomer specific or can it be used for other monomers? Does this method can be used to prepared copolymers?
2. Though active monomer mechanism is quite convincing, the scheme that authors depicted in Figure 3a is confusing. Can the terminal carboxylate attack the C5-carbonyl in NCA? If this is the case, the intramolecular cyclization will be possible. Or it should be a proton exchange step during the terminal carboxylate and 3-NH in NCA happened first?

Reviewer #2 (Remarks to the Author):

This manuscript reports the superfast preparation of synthetic polypeptide polymers using "purely" anionic ring-opening polymerization (ROP) of alpha-amino acid N-carboxyanhydride (NCA). Overall, the work is comprehensively described and the manuscript may represent a significant step further for the scientific community in the preparation and application of synthetic polypeptides polymers. All in all, this manuscript contains some interesting observations, but substantial experimental work is necessary to demonstrate how (or until where) the proposed catalyst is superior in practice to known ROP of NCA (see my comments below).

I have three main concerns regarding the work submitted by Runhui Liu and coworkers:

- 1) My major concern relies on the following statement of the authors:

"It is noteworthy that when using primary amine as the initiator to prepare around 20 mer polymers, the polymerization of BLG-NCA and Lys-NCA both encountered puzzling bimodal GPC peaks (Fig 2d, 2e) similar to the observations reported in precedent literature articles by Dr. Cheng, Dr. Chan-Park and Dr. Barz individually. However, bimodal GPC trace was not observed in LiHMDS or HMDS initiated polymerization of BLG-NCA and Lys-NCA." It is, at that point, important

to consider that this bimodal peak involves the propensity of polypeptides to form secondary structures (alpha helix or beta sheets, see ref 62 of the manuscript for instance). This structuring ability of polypeptide polymers is an important feature for materials scientists to prepare nanomaterials, gels, membranes or surface coatings. One key requirement for secondary structuring is to preserve the chirality of the CH carbons belonging to the main backbone during polypeptide preparation. It is important to note that racemization of the backbone may significantly decrease the structuring ability (see for instance Vicent and coworkers, *Polym. Chem.*, 2013, 4, 3182). Considering the experimental observations presented by the authors (SEC traces), I strongly suspect that the proposed anionic mechanism induces significant racemization of the polypeptide backbone. Whereas it is not necessarily important to keep the chirality of the amino acids for some biomedical applications, this point could be critical for some other applications. Authors might therefore provide an evaluation of this racemization process to strength this article. According to the paper of Vicent (*Polym. Chem.*, 2013, 4, 3182), deprotection of PBLG to afford PGA may permit to access this evaluation (polarimetry). I would like to point out that in the paper of Vicent, values are indeed not specific rotations but measured rotations. Specific rotations of PGAs are comprised between -80 and -90 deg.dm⁻¹.g⁻¹.mL when the deprotection step is correctly performed.

2) My second concern is related to the stability of NCA monomers in both THF and DMF. The results presented by the authors are critical, particularly when they handle BLG-NCA in DMF. I might accept that the monomers are quite sensitive but my experience on this topic prompt me to consider that these results are exaggerated. I would therefore recommend, in each case, to measure the water content in ppm (Karl Fisher) to nuance the given results by taking into account the water content. I strongly suspect that the differences observed between DMF and THF comes from this parameter (usually, commercial anhydrous DMF contains more than 200 ppm of water whereas anhydrous THF contains less than 100 ppm of water).

3) My last concern is related to the end extremities obtained upon LiHMDS initiated ROP. After carefully reading the document, it is not clear whether this end extremity could be tuned or not. In this context, the following reference should be seriously taken into account: *Biomacromolecules*, 2016, 17, 891. Moreover, the proposed LiHMDS process should be nuanced when compared to previous synthetic methods that permit to introduce, easily, a wild scope of amine initiators (this includes an easy access to surface initiated ROP). I would add, here, a specific comment related to PBLG, one of both polymers prepared by authors: in this specific case and, as it is well documented in ref 47 of the manuscript, cyclisation at the end extremity might occur during the ROP process. I would believe that 1) during propagation, the anionic process enhances the probability of this side reaction; 2) during initiation, upon LiHMDS deprotonation, the anionic NCA monomer could therefore attack the NCA moiety of another monomer but also the benzyl ester of the side chain.

Reviewer #3 (Remarks to the Author):

This manuscript describes use of LiHMDS as a polymerization initiator for NCAs. Polymerization of BLG-NCA and Boc-Lys NCA using LHMDS is compared to hmds and hexylamine initiators. The LiHMDS method was used to make copolymers of PBLG and Lys which were examined for antimicrobial properties.

The main innovation in this work appears to be that LiHMDS is a bench stable initiator for NCA polymerization, which is certainly a useful property for any chemical procedure. Other purported benefits of this system are rapid speed and possibility of high MW chains of ca 500 residues. However, these qualities are not new or unique to this system. More than 20 years ago, Deming reported Ni, Co, and Fe initiators that polymerize NCAs at essentially the same time scale as LHMDS, and can yield even higher MW polypeptides on multi-gram scale. Deming's initiators do require an oxygen free and anhydrous environment, necessitating use of a glovebox. A bench stable initiator would certainly be a convenient option, however, NCA monomers themselves are

subject to rapid hydrolysis should be stored in a glovebox. If the chemist needs to use a glovebox for the monomer, he/she may not be particularly burdened by use of a metal catalyst in the same location. Ultimately, LiHMDS should be compared side-by-side to transition metal catalysts rather than hexylamine. Surprisingly, the metal catalysts are barely mentioned in this work.

Deming, T. J. *Nature*, 1997, 390, 386-389.

A few particular issues that need further experiments and work include:

1. The stability of NCAs in THF/DMF is not a particularly interesting addition to the manuscript. The data is not informative since the ppm water is not characterized or standardized. Solvents should be analyzed by a Karl Fischer titrator. The results reported for NCA stability in THF vs DMF probably don't indicate stability in these solvents, but rather, may just be a canary in the coal mine for wet DMF. Further, chloride ions are known initiators in DMF so NCA purity may also be a factor. The SI says anhydrous solvents were purchased and used without further purification, or were dried over MgSO₄, or distilled (for THF). The water content of 'anhydrous' solvents varies widely by manufacturer, by lot number, and in laboratory handling. Distillation is subject to the technique of the chemist and can result in THF of different ppm water, and MgSO₄ is only a crude method of drying. DMF is particularly challenging to dry. Further, depending on how the DMF was purified it may be contaminated with dimethylamine, which can initiate NCA polymerization. So overall, these data are not thoroughly examined for all factors and should be removed from the paper.

2. The devil may also be in the details of commercial LiHMDS. There are two processes used to prepare the reagent, via n-BuLi or Li metal, and LiHMDS from different suppliers may be prepared by different methods and polymerization results may vary. LiHMS quality has troubled Pfizer process chemists. The authors should verify that LiHMDS from different sources or different lots give similar results.

[dx.doi.org/10.1021/op4002356](https://doi.org/10.1021/op4002356)

[dx.doi.org/10.1021/op400236r](https://doi.org/10.1021/op400236r)

[dx.doi.org/10.1021/op400237j](https://doi.org/10.1021/op400237j)

3. LiHMDS polymerization of Boc-L-Lys NCA is reported to take 3.5-10x longer to complete than BLG-NCA. It is strange that the kinetics of the two are so different and the authors should investigate this further. The kinetics put into question the utility of the initiator for construction of the proposed polypeptide libraries. If the kinetics are so different for various NCAs, the initiator would not be useful for statistical copolymers. The antimicrobial polypeptides reported in Fig 4 are more likely to be tapered polymers than statistical mixtures as shown.

Items for correction or clarification:

1. What is the mobile phase for the HPLC used to determine stability of NCAs and conversion to polymer? If this is not run in anhydrous aprotic solvents, the data will be skewed by NCA hydrolysis and polymerization.

2. About 10 years ago IUPAC recommended discontinuing the use of the misleading phrase "Polydispersity index" and "PDI". Dispersity index should be used instead.

3. Lys-NCA should be called Boc-L-Lys-NCA if the protecting group for Glu NCA is noted. Naming should be standardized.

4. Table 1 should have m:I ratios

5. Fig 1c, Boc-L-Lys NCA is missing stereochemistry

6. The term 'deprotonating agent' could be better stated as a 'base'

7. GPC traces are truncated in the figures which could obscure information. They should be shown in full in the SI if there's not room in the main text.

8. There are a considerable number of typos and convoluted language. The authors are advised to seek a native English speaker for editing service

We greatly thank all the reviewers for the comments and suggestions. These suggestions for modification are very helpful, and the resulting changes have significantly improved the manuscript.

In recognizing all reviewers' helpful comments and suggestions, we thank all reviewers in the acknowledgement of the revised manuscript.

Our responses to reviewer comments are detailed below; for reference, the reviewer comments/questions are presented in *italics*, with our response following.

Reviewer 1

Comments: In this manuscript, Dr Liu and his co-workers reported an easy handled and superfast ring opening polymerization of NCA by using LiHMDS as initiator. This is a quite interesting research direction, considering the wide application of the peptidomimics as functional biomaterials. The authors claim they developed a LiHMDS initiated superfast NCA polymerization that can finish within a range of mins to hours and this polymerization method is tolerant to moisture, so that it can be operated successfully in open vessels outside of a glovebox without any protection. They have synthesized up to 500 DP of polypeptides with low polydispersity (PDI=1.15-1.26) and they also demonstrated a successful open-flask polymerization using this method, obtaining 1.47g polymers. Though the results are promising, the fast polymerization of NCA has been illustrated quite long time ago using the organonickel compounds (NATURE | VOL 390 | 27 NOVEMBER 1997). The rate constant is $2.7 \times 10^{-4} \text{ s}^{-1}$ (0.67 mM initiator) compared to 1.7 h^{-1} (2 mM initiator). The polymerization speed of these two methods is at least comparable. But considering the toxicity issue of transition metal used, the finding of LiHMDS as initiator is still quite interesting, especial for preparing functional biomaterials. The most important finding is the high tolerant polymerization method and it can be carried outside of glovebox. It will be much easier to carry out the NCA polymerization by this method. It will be a useful new NCA polymerization technique and will be interesting to a broad field, from polymer chemistry, material and biological science.

However, there are some limitations that the authors should be addressed before its further consideration for publication.

Response: We appreciate the favorable comments from the reviewer. We believe two of the advantages for the LiHMDS initiated NCA polymerization will be attractive to researchers in broad fields as mentioned by the reviewer in above general comments.

Firstly, the LiHMDS initiated NCA polymerization can be operated open vessel without any protection outside of the glovebox. This NCA polymerization is easy to handle especially for library synthesis and large scale synthesis. Therefore, this new NCA polymerization method will be interesting to researchers from broad fields including but not limited to polymer chemistry, biomaterials and biological science.

Secondly, primary amine initiated NCA polymerization is the most widely used method to prepare polypeptide polymers. This reaction normally takes 2-3 days or even longer, so there is a concern on polymerization of NCA monomers that are unstable. Therefore, the fast NCA polymerization initiated by LiHMDS is highly favorable especially for NCA monomers that are quite unstable. It is also pointed out by the reviewer that organonickel compounds (NATURE | VOL 390 | 27 NOVEMBER 1997) can also catalyze fast NCA polymerization, but the use of transition metal catalyst is associated with toxicity issue and concerns especially for studies in biomaterials and biological science. It worth mentioning that the transition metal catalyst reported in the Nature paper was not commercially available. Since this catalyst is very sensitive to both moisture and oxygen, it is also a great challenge to synthesize the transition metal catalyst for researchers who are not well trained in organic chemistry and organometallic chemistry. In fact, more and more researchers are exploring the functions and applications of polypeptides and many of these researchers have backgrounds other than organic chemistry and organometallic chemistry. This could be one of the reasons why primary amine initiated NCA polymerization is still the most widely used method to prepare polypeptides. One of the merits of our method is that LiHMDS is a common and commercially available chemical. We tested LiHMDS from different commercial sources and all initiate NCA polymerization successfully to give similar results.

All specific questions and suggestions from the reviewer were responded below in details.

1. The open-flask NCA polymerization is very impressive, but it was only conducted to synthesize a low molecular weight of PBLG. It is highly advised that the author can conduct a

more complete study of this moisture tolerant NCA polymerization method. For example, what is the highest DP of polymers that can be prepared via this method? Can the polymer chain still be further extended when conducted outside the glovebox? Does this polymerization method is monomer specific or can it be used for other monomers? Does this method can be used to prepared copolymers?

Response: We thank the reviewer for these valuable suggestions to help us improve our manuscript with more detailed study on LiHMDS initiated open-flask NCA polymerization. By following the suggestions from the reviewers, we did more complete studies as described below.

We studied the open-flask polymerization of BLG NCA and successfully obtained corresponding polymers poly-BLG with variable DP (30, 143, 404, 699, and 1294) and narrow dispersity index (M_w/M_n in the range of 1.08-1.28). In these studies, we can get poly-BLG with a highest DP up to 1294 ($M_n = 283400$). These data were summarized in the new Table 3 of the revised manuscript (entry 1-5) of the modified manuscript. We also added description of these new data to the main text of revised manuscript accordingly.

To explore if the polymer chain can still be further extended when the NCA polymerization is conducted open vessel outside of the glovebox, we did test using Boc-L-Lys NCA by adding the monomer in two portions. The result indicated that the first block of poly-Boc-L-Lys is easily obtained with DP=27 and $M_w/M_n=1.23$, and the addition of extra Boc-L-Lys NCA to the reaction led to successful extension of the polymer to give a final block copolymer with DP=61 and $M_w/M_n=1.29$. These data were included into the new Table 3 of the revised manuscript (entry 15). We also added description of this new result into the main text of revised manuscript accordingly.

To demonstrate that the open vessel NCA polymerization initiated by LiHMDS outside of the glovebox is not limited to just BLG NCA, we explored this polymerization method using several other NCA monomers including Boc-L-Lys NCA, Boc-L-Orn NCA, tBu-L-Ser NCA, L-Leu NCA, and tBu-L-Asp NCA. The polymerization of Boc-L-Lys NCA successfully afforded corresponding polymers poly-Boc-L-lys at variable DP (30, 59, 275, and 486) with M_w/M_n in a range of 1.18-1.25 as shown in the new Table 3 of the revised manuscript (entry 6-9). Poly-Boc-L-lys at high DP has limited solubility in DMF and this solubility issue limited the GPC characterization of resulting poly-Boc-L-lys with a polymer length above 500 mers. The

polymerization of Boc-L-Orn NCA also proceeded successfully to give the poly-Boc-L-Orn. We demonstrated the polymerization of Boc-L-Orn NCA and obtained poly-Boc-L-Orn at DP=28 ($M_w/M_n=1.23$) and DP=55 ($M_w/M_n=1.24$) respectively as shown in the new Table 3 of the revised manuscript (entry 10-11). Poly-Boc-L-Orn at higher polymer length has low solubility in DMF and THF. These results indicate that LiHMDS initiated open-vessel polymerization of NCA is suitable to a wide range of NCA monomers. We also tested other NCA monomers for their copolymerization using 1:1 mixture of two NCA monomers ((Boc-L-Lys NCA + tBu-L-Ser NCA), (Boc-L-Orn NCA + L-Leu NCA), and (BLG NCA + tBu-L-Asp NCA)), initiated by LiHMDS in open flasks outside of the glovebox. We obtained all copolymers successfully with DP=41, 39, and 35 respectively and narrow M_w/M_n at 1.24-1.25 as summarized in the new Table 3 of the modified manuscript (entry 12-14) of the modified manuscript. We also added description of these new results to the main text of revised manuscript accordingly.

2. Though active monomer mechanism is quite convincing, the scheme that authors depicted in Figure 3a is confusing. Can the terminal carboxylate attack the C5-carbonyl in NCA? If this is the case, the intramolecular cyclization will be possible. Or it should be a proton exchange step during the terminal carboxylate and 3-NH in NCA happened first?

Response: We thank the reviewer for comments on the polymerization mechanism and these comments inspired us to further clarify our description and explanation on this part. First, the terminal lithium carboxylate (as part of the terminal carbamate group) in updated Figure 4a of the revised manuscript (Figure 3a in our original manuscript) is a reasonable nucleophile to attack an electrophile in general organic reaction. So, it is reasonable to propose that the terminal lithium carboxylate in Figure 4a of the revised manuscript can attack the C5-carbonyl in NCA for chain propagation. Our hypothesis was supported by precedent literature confirming that a terminal carbamate can attack the C5-carbonyl in NCA and lead to the ring opening polymerization of NCA (*Journal of Polymer Science: Polymer Letters Edition*, 1978, 16(10):491–496.). In addition, we also get support on this hypothesis from our new Mass Spectrometer data as shown in Figure 4 of the revised manuscript (4c for proposed possible reaction of terminal carbamate group with NCA; 4e for corresponding Mass Spec result). Taking the Boc-D,L-Lys NCA for example, the initiation step by LiHMDS resulted in a terminal

carbamate intermediate 4 that may continue to react with NCA monomers via two possible routes as depicted in Figure 4c of the revised manuscript. As we proposed for the chain propagation of LiHMDS initiated NCA polymerization, the terminal carboxylate in intermediate 4 can attack the C5-carbonyl in another NCA directly to open the NCA ring. This process will add one extra amino acid into the polypeptide chain to give compound 6 as a tripeptide. Such a hypothesis was supported by the finding of an obvious peak of tripeptide 6 at m/z 779.4641 in the HR ESI-MS spectrum (Figure 4e in the revised manuscript). In addition, the MALDI-TOF MS characterization on purified polypeptide also confirmed the presence of an N-terminal lithium carboxylate group (Supplementary Figure 7a in the revised manuscript). Another possible reaction route is that the terminal carboxylate in intermediate 4 has proton exchange with 3-NH in NCA, as also wondered by the reviewer, to generate an active NCA monomer and an N-terminal carboxylic acid (as part of the protonated carbamate). Then the active NCA can undergo nucleophilic addition to the C5-carbonyl in NCA group at the C-terminus of the dipeptide to afford a tripeptide intermediate that loses one molecule of CO_2 , during purification operation, to give a tripeptide 7 with N-terminal NH_2 (Figure 4c in the revised manuscript). However, such a tripeptide 7 was not found in corresponding HR ESI-MS spectrum (Figure 4e in the revised manuscript). This result implies that the proton exchange route is neglectable though we can't totally rule out this possibility. This result is consistent to the organic chemistry that the terminal carboxylate (as part of the carbamate) is an even weaker base than the 3-NH in NCA and thus the deprotonation of 3-NH in NCA by terminal carboxylate is not favored. Therefore, the first route of terminal lithium carboxylate (as part of the carbamate group) attacking C5-carbonyl in a new NCA plays a dominate role during the polymer chain propagation. We also included related discussions into the revised manuscript accordingly.

For the reviewer's concern about side reaction of intramolecular cyclization, theoretically it is possible to happen since the N-terminal carboxylate can perform as a nucleophile to attack C-terminal NCA in the same chain during polymerization. It is also reported in the literature (*Macromolecules* 2005, 38, 5513-5518) that intermolecular cyclization during NCA polymerization can be easily found in MALDI-TOF MS of the resulting polypeptide. We carefully checked the MALDI-TOF MS of the obtained poly-BLG from LiHMDS initiated polymerization and found no signal of cyclized polypeptide (Supplementary Figure 7a in the revised manuscript). We also got solid support from our terminal functionalization study on Boc-

D,L-Lys NCA polymerization (Figure 4h in the revised manuscript). In this study, we did LiHMDS initiated polymerization and prepared poly-Boc-D,L-Lys at 20 mer. At the end of polymerization, addition of 4-tert-butylbenzylamine to the reaction mixture afforded polymers with expected C-terminal 4-tert-butylbenzyl group (Figure 4h in the revised manuscript). To further tune the N-terminal end group, we did a demonstration by mixing above polymer with an NHS-ester terminated maleimide (N-succinimidyl 3-maleimidopropionate) to obtained polypeptides with an N-terminal maleimide group (Figure 4h in the revised manuscript). Proton NMR characterization obtained an integration ratio of 2:2:20 for H_b (C-terminal Benzyl): H_i (N-terminal maleimide): H_a (backbone CH), which indicates a successful and controlled terminal functionalization on both ends. The resulting polypeptide has almost one C-terminal 4-tert-butylbenzyl group and one N-terminal maleimide group per polypeptide chain (Figure 4g in the revised manuscript). This result also supports aforementioned statement that acyclic polymers dominate in LiHMDS initiated NCA polymerization, though we can't totally rule out the possibility in forming cyclic polypeptides. We also included above data and discussions into the revised manuscript accordingly.

Reviewer: 2

Comments: *This manuscript reports the superfast preparation of synthetic polypeptide polymers using "purely" anionic ring-opening polymerization (ROP) of alpha-amino acid N-carboxyanhydride (NCA). Overall, the work is comprehensively described and the manuscript may represent a significant step further for the scientific community in the preparation and application of synthetic polypeptides polymers. All in all, this manuscript contains some interesting observations, but substantial experimental work is necessary to demonstrate how (or until where) the proposed catalyst is superior in practice to known ROP of NCA (see my comments below). I have three main concerns regarding the work submitted by Runhui Liu and coworkers:.*

Response: We thank the reviewer for above general comments and specific questions/suggestions below.

1. My major concern relies on the following statement of the authors: "It is noteworthy that when using primary amine as the initiator to prepare around 20 mer polymers, the polymerization of BLG-NCA and Lys-NCA both encountered puzzling bimodal GPC peaks (Fig 2d, 2e) similar to the observations reported in precedent literature articles by Dr. Cheng, Dr. Chan-Park and Dr. Barz individually. However, bimodal GPC trace was not observed in LiHMDS or HMDS initiated polymerization of BLG-NCA and Lys-NCA." It is, at that point, important to consider that this bimodal peak involves the propensity of polypeptides to form secondary structures (alpha helix or beta sheets, see ref 62 of the manuscript for instance). This structuring ability of polypeptide polymers is an important feature for materials scientists to prepare nanomaterials, gels, membranes or surface coatings. One key requirement for secondary structuring is to preserve the chirality of the CH carbons belonging to the main backbone during polypeptide preparation. It is important to note that racemization of the backbone may significantly decrease the structuring ability (see for instance Vicent and coworkers, *Polym. Chem.*, 2013, 4, 3182). Considering the experimental observations presented by the authors (SEC traces), I strongly suspect that the proposed anionic mechanism induces significant racemization of the polypeptide backbone. Whereas it is not necessarily important to keep the chirality of the amino acids for some biomedical applications, this point could be critical for some other applications. Authors might therefore provide an evaluation of this racemization process to strength this article. According to the paper of Vicent (*Polym. Chem.*, 2013, 4, 3182), deprotection of PBLG to afford PGA may permit to access this evaluation (polarimetry). I would like to point out that in the paper of Vicent, values are indeed not specific rotations but measured rotations. Specific rotations of PGAs are comprised between -80 and -90 deg.dm⁻¹.g⁻¹.mL when the deprotection step is correctly performed..

Response: We thank the reviewer for raising this important question about possible racemization of the polypeptide backbone during polymerization and providing hints for our further exploration on this. As mentioned by the reviewer, we said in our original manuscript "It is noteworthy that when using primary amine as the initiator to prepare around 20 mer polymers, the polymerization of BLG-NCA and Lys-NCA both encountered puzzling bimodal GPC peaks.... However, bimodal GPC trace was not observed in LiHMDS or HMDS initiated polymerization of BLG-NCA and Lys-NCA." In our original manuscript, the "Lys-NCA" was a

racemic mixture *N* ϵ -*tert*-butyloxycarbonyl- α -D/L-lysine NCA. We used “Lys-NCA” as the abbreviation of the racemic mixture *N* ϵ -*tert*-butyloxycarbonyl- α -D/L-lysine NCA, which was described in the third sentence of paragraph 2 in our original manuscript. Although our purpose was to simplify the name of NCA and make the manuscript concise, obviously our abbreviation for the *N* ϵ -*tert*-butyloxycarbonyl- α -D/L-lysine NCA (more accurately should be Boc-D,L-Lys NCA rather than previously used Lys-NCA) led to misunderstanding. We observed that using primary amine as the polymerization initiator, around 20 mer polymers from BLG NCA and Boc-D,L-Lys NCA both encountered puzzling bimodal GPC peaks. Because we used the Boc-D,L-Lys NCA, a racemic mixture rather than a homochiral NCA, the resulting polymer poly-Boc-D,L-Lys is not likely to have secondary structures like alpha helix and beta sheets. That’s why we described our observation as “puzzling bimodal GPC peaks”. It is noteworthy that we didn’t observe this bimodal GPC peaks on polypeptides prepared from LiHMDS initiated NCA polymerization. In addition to above explanation, we didn’t observed such puzzling bimodal GPC peaks on polypeptides with a longer chain length (40 mer and above) when using primary amine to initiate the polymerization of BLG NCA and Boc-D,L-Lys NCA. This observation also implies that there is likely no correlation between primary amine initiator resulted bimodal GPC peaks and secondary structure propensity of resulting polypeptides. To clarify our abbreviation on NCA monomers and avoid aforementioned misunderstanding, we modified our abbreviation on the racemic mixture *N* ϵ -*tert*-butyloxycarbonyl- α -D/L-lysine NCA as Boc-D,L-Lys NCA and made changes accordingly in the revised manuscript. We deeply thank the reviewer for this question to help us optimize our NCA abbreviation and clarify the description on this very important part.

We recognize that the reviewer raised an important and interesting question to think about “if the proposed anionic mechanism induces significant racemization of the polypeptide backbone” as mentioned in the reviewer’s comments. We then followed the reviewer’s suggestion to evaluate the possible racemization of poly-BLG that was derived from homochiral BLG NCA and can have helical structures. Firstly, we used circular dichroism (CD) spectroscopy to characterize poly-BLG that was obtained from LiHMDS initiated polymerization of BLG NCA. The solutions of poly-BLG at variable chain length (34, 74 and 124 mers) in hexafluoroisopropanol (HFIP) all displayed a positive band at 195 nm and two negative bands at 208 nm and 220 nm, which are characteristic bands of α -helix (please see CD spectrum in Figure

2c of the revised manuscript). Secondly, we followed the reviewer's suggestion and the reference (Polym. Chem., 2013, 4, 3182), and measured the specific rotations of deprotected poly-BLG by polarimetry. We measured three batches of deprotected poly-BLG and obtained specific rotation values in the range of -78 to $-83 \text{ deg}\cdot\text{dm}^{-1}\cdot\text{g}^{-1}\cdot\text{mL}$ that falls into the expected specific rotation range of poly-BLG. In addition, to mimic polypeptides with variable degree of backbone racemization, we also synthesized polypeptides from a mixture of γ -benzyl-L-glutamate NCA (BLG NCA) and γ -benzyl-D-glutamate NCA (BDG NCA) by increasing the content of BDG NCA incrementally from 0% to 50% of the total NCA mixture. As shown in the Supplementary Figure 6 of the revised manuscript, there is linear relationship between the specific rotation (the Y axis) and the percentage of BLG subunit (L-configuration from 50% to 100%, the X axis) that mimics the degree (percentage) of backbone racemization. According to this linear fitting, it is clear that the specific rotation of this polypeptide is very sensitive to racemization of the backbone CH carbon. As aforementioned, three batches of polypeptides were synthesized by LiHMDS initiated polymerization of BLG and the resulting polypeptides were deprotected to have specific rotation in the range of -78 to $-83 \text{ deg}\cdot\text{dm}^{-1}\cdot\text{g}^{-1}\cdot\text{mL}$ that falls into the expected specific rotation range of poly-BLG at round $-80 \text{ deg}\cdot\text{dm}^{-1}\cdot\text{g}^{-1}\cdot\text{mL}$. These two set of data altogether support that racemization of the backbone in LiHMDS initiated NCA polymerization is not significant, even though we can't totally rule out the possibility. This conclusion is also consistent to that based on analysis of CD spectrum aforementioned. We have included these results and related discussions in the revised manuscript.

2. My second concern is related to the stability of NCA monomers in both THF and DMF. The results presented by the authors are critical, particularly when they handle BLG-NCA in DMF. I might accept that the monomers are quite sensitive but my experience on this topic prompt me to consider that these results are exaggerated. I would therefore recommend, in each case, to measure the water content in ppm (Karl Fisher) to nuance the given results by taking into account the water content. I strongly suspect that the differences observed between DMF and THF comes from this parameter (usually, commercial anhydrous DMF contains more than 200 ppm of water whereas anhydrous THF contains less than 100 ppm of water).

Response: We thank the reviewer for this important question and inspiring comments. Both THF and DMF were purchased from Sigma-Aldrich as anhydrous solvents and used directly in the glove box for NCA stability test. We followed the suggestion from the reviewer and carefully measured the water content of anhydrous DMF and THF using a Karl Fisher titrator. The actual water content of our DMF and THF was found to be 80 ppm and 46 ppm respectively that are slightly higher than the numbers in the quality control report (20 ppm and 10 ppm water content respectively for DMF and THF) from Sigma-Aldrich. We agree with the reviewer that moisture in the solvent can contribute to the loss of NCA and the difference of NCA stability in different solvent, THF vs. DMF. It is also possible that the water content of anhydrous solvents vary in different solvent providers and different lots from the same provider. Different labs are also likely to have variable levels of facilities and care in handling anhydrous solvent. In light of this observation and consideration, we moved the NCA stability content from Figure 1a, 1b in the main text of our original manuscript to the supporting information (Supplementary Figure 1) of our revised manuscript. We did this adjustment to avoid confusion but still provide this critical information about NCA stability for readers. These results and related discussions were included into the main text and supplementary information of our revised manuscript.

3. My last concern is related to the end extremities obtained upon LiHMDS initiated ROP. After carefully reading the document, it is not clear whether this end extremity could be tuned or not. In this context, the following reference should be seriously taken into account: Biomacromolecules, 2016, 17, 891. Moreover, the proposed LiHMDS process should be nuanced when compared to previous synthetic methods that permit to introduce, easily, a wide scope of amine initiators (this includes an easy access to surface initiated ROP). I would add, here, a specific comment related to PBLG, one of both polymers prepared by authors: in this specific case and, as it is well documented in ref 47 of the manuscript, cyclisation at the end extremity might occur during the ROP process. I would believe that 1) during propagation, the anionic process enhances the probability of this side reaction; 2) during initiation, upon LiHMDS deprotonation, the anionic NCA monomer could therefore attack the NCA moiety of another monomer but also the benzyl ester of the side chain.

Response: We thank the reviewer for the comments and the important and specific questions at the end of the comments. According to our proposed mechanism of LiHMDS initiated NCA polymerization, the resulting polypeptides have a C-terminal active NCA moiety and an N-terminal carboxylate group that enable easy functionalization or turning of both end groups. The C-terminal NCA subunit can be attacked by a nucleophile, like a primary amine, to tune the end group conveniently at the C-termini of the polypeptides. The N-terminal carboxylate group can react with an electrophile to introduce variable functional groups at the N-termini of the polypeptides. To make demonstration on these, we did extra experiments using the polymerization of Boc-D,L-Lys NCA as an example as shown in Figure 4 of the revised manuscript (Figure 4h for the reaction scheme; 4g for evidence of proton NMR characterization). At the end of the normal operation for LiHMDS initiated polymerization of Boc-D,L-Lys NCA, we added 4-*tert*-butylbenzylamine as a nucleophilic reagent to the reaction mixture and obtained polymers with expected C-terminal 4-*tert*-butylbenzyl group. Proton NMR characterization confirmed that we got almost one C-terminal 4-*tert*-butylbenzyl group per polypeptide chain for this 20 mer polypeptide, which means a successful and controlled C-terminal functionalization. To further tune the N-terminal end group, we did a demonstration by mixing above purified polymer with a NHS ester functionalized maleimide (N-succinimidyl 3-maleimidopropionate) and obtained polypeptide with an N-terminal maleimide group. The proton NMR of deprotected polypeptide confirmed that we got almost one N-terminal maleimide group per polymer chain. Proton NMR characterization obtained an integration ratio of 2:2:20 for H_b (C-terminal Benzyl): H_l (N-terminal maleimide): H_a (backbone CH), which indicates a successful and controlled terminal functionalization on both ends (Figure 4g in the revised manuscript). These results demonstrated that LiHMDS initiated NCA polymerization can provide convenient and efficient functionalization at either one end (N-termini or C-termini of the polypeptides) or both ends. We also cited the paper (*Biomacromolecules*, 2016, 17, 891) mentioned in above comments by the reviewer as ref. 38 in our original and revised manuscripts. These results and related discussions were included into the revised manuscript.

The reviewer also raised an interesting question about possible cyclization at the end extremity as a side reaction during propagation of BLG NCA polymerization initiated by LiHMDS. We then investigated this issue by functionalizing the synthesized BLG polymer at the N-terminus with an electrophile to reaction with N-terminal carboxylate. As shown in

Supplementary Figure 38 of the revised manuscript, proton integration in ^1H NMR give a ratio at almost 2:70:35 for $\text{H}_k(\text{N-terminal maleimide}):\text{H}_d(\text{sidechain benzyl}):\text{H}_a(\text{backbone CH})$, which means almost one N-terminal maleimide group was attached to one poly-BLG chain. This observation indicates that cyclization at the end extremity, as a possible side reaction during propagation of BLG NCA polymerization initiated by LiHMDS, is not significant in our case. The reviewer also suspected that “during initiation, upon LiHMDS deprotonation, the anionic NCA monomer could therefore attack the NCA moiety of another monomer but also the benzyl ester of the side chain”. To check this possibility, we analyzed the proton NMR spectrum of poly-BLG obtained from LiHMDS initiated polymerization. An equal molar ratio between benzyl ester and BLG backbone was observed based on the integration of protons in ^1H NMR as shown in the Supplementary Figure 7b of revised manuscript. The NMR data indicated that benzyl ester of the side chain is retained in the resulting poly-BLG. This experimental result agrees with our expectation that the anionic NCA monomer selectively attacks another NCA monomer rather than the side chain benzyl ester. This result is not too surprising since the anhydride carbonyl group is much more reactive than benzyl ester carbonyl group for nucleophilic addition. We also included these results and related discussions into the revised manuscript accordingly.

Reviewer: 3

Comments: *This manuscript describes use of LiHMDS as a polymerization initiator for NCAs. Polymerization of BLG-NCA and Boc-Lys NCA using LHMDS is compared to hmds and hexylamine initiators. The LiHMDS method was used to make copolymers of PBLG and Lys which were examined for antimicrobial properties.*

The main innovation in this work appears to be that LiHMDS is a bench stable initiator for NCA polymerization, which is certainly a useful property for any chemical procedure. Other purported benefits of this system are rapid speed and possibility of high MW chains of ca 500 residues. However, these qualities are not new or unique to this system. More than 20 years ago, Deming reported Ni, Co, and Fe initiators that polymerize NCAs at essentially the same time scale as LHMDS, and can yield even higher MW polypeptides on multi-gram scale. Deming's

initiators do require an oxygen free and anhydrous environment, necessitating use of a glovebox. A bench stable initiator would certainly be a convenient option, however, NCA monomers themselves are subject to rapid hydrolysis should be stored in a glovebox. If the chemist needs to use a glovebox for the monomer, he/she may not be particularly burdened by use of a metal catalyst in the same location. Ultimately, LiHMDS should be compared side-by-side to transition metal catalysts rather than hexylamine. Surprisingly, the metal catalysts are barely mentioned in this work. (Deming, T. J. Nature, 1997, 390, 386-389.) A few particular issues that need further experiments and work include:

Response: We thank the reviewer for the general comments and suggestions. What encourages us to explore new NCA polymerization chemistry is that conventional NCA polymerization strategies, like primary amine initiated NCA polymerization and transition metal catalyzed NCA polymerization, are very sensitive to moisture. Therefore, there are stringent requirements on dry solvent and dry environment like operating in a glovebox. Operating in the glove box makes it inconvenient for library synthesis. Moreover, NCA polymerization and resulting polypeptides have attracted more and more attentions and interests from researchers whose have background in biology, materials, drug delivery, and tissue engineering rather than organic chemistry. Many of these researchers would like to utilize NCA polymerization, but they may not have a glovebox in their labs. So, it is challenging for many researchers to run NCA polymerizations that have strict requirement in dry conditions or need special skills in organic or organometallic chemistry to prepare the catalyst, like the transition metal catalyst mentioned by the reviewer in the *Nature* paper (the bipyNi(COD) catalyst in this paper is not commercially available). For these reasons, we contribute the LiHMDS initiated NCA polymerization to the scientific community because this new NCA polymerization chemistry has five major advantages: 1) amenable to open vessel reaction outside of the glovebox to facilitate parallel synthesis of polypeptide libraries, 2) fast polymerization for efficient polypeptide synthesis and suppression of side reactions, 3) suitable to prepare long chain polypeptides with DP up to 1294 for poly-BLG ($M_n=283400$), 4) stable and commercially available LiHMDS as the initiator for NCA polymerization, and 5) be free from the problem and concerns of cytotoxicity that is always associated with transition metal catalyst especially for application in biomaterials and biological science as mentioned by one of the reviewers.

In the first paragraph of our original manuscript, when introducing progress of NCA polymerization chemistry we mentioned transition metal catalysts by Prof. Deming first because this is a very important contribution to this field. In our original manuscript, we also cited the *Nature* paper about transition metal catalysts (ref 34 in our original manuscript) mentioned by the reviewer in above comments (Deming, T. J. *Nature*, 1997, 390, 386-389.). We compared our LiHMDS method with primary amine method majorly because primary amine is the most commonly used initiator for NCA polymerization and polypeptide synthesis in literatures. Nonetheless, we agree with the reviewer that side-by-side comparison with the catalyst in the *Nature* paper is helpful and we followed the request from the reviewer to compare these initiators in identical reaction conditions in THF regarding the conversion of NCA, characterizations of resulting polypeptides, and reaction rate. As summarized in Figure 1b and 1c of the revised manuscript, the transition metal catalyst bipyNi(COD) can initiate fast NCA polymerization to prepare moderate MW polypeptide in THF. However, when preparing large MW polypeptides the conversion of NCA by transition metal catalyst was much lower than did LiHMDS. For polymerization of BLG NCA, the transition metal catalyst bipyNi(COD) can generate moderate to high MW polypeptides with narrow dispersity index (Table 1, entry 12-14 in the revised manuscript), comparable to the result obtained from LiHMDS initiated polymerization. Then, we examine the transition metal catalyst on Boc-D,L-Lys NCA that has low reactivity due to its racemic nature (Table 2, entry 12-14 in the revised manuscript). Using transition metal catalyst bipyNi(COD) can only get moderate MW polypeptide with DP up to 75 in THF. Surprisingly, the attempt to prepare 500 mer poly-Boc-D,L-Lys using bipyNi(COD) as the initiator obtained only short polypeptide (DP=25) with most of the NCA left over even after the reaction continued for 5 days. In comparison, LiHMDS can initiate the polymerization of Boc-D,L-Lys NCA to give poly-Boc-D,L-Lys that have high MW with DP up to 405 (Table 2, entry 5 in the revised manuscript). To have the side-by-side comparison on the reaction rate initiated by transition metal catalyst bipyNi(COD) and LiHMDS, we did extra experiments on polymerization of BLG NCA using bipyNi(COD) as the initiator and update the data in Supplementary Figure 5a of the revised manuscript. The transition metal catalyst bipyNi(COD) initiated BLG NCA polymerization has a rate constant of 0.87 h^{-1} that is half of the rate constant for LiHMDS initiated BLG NCA polymerization, when both polymerizations were conducted under identical conditions in THF. In new Table 3 of the revised manuscript, we also

demonstrated successful polymerization of variable NCA monomers via LiHMDS initiated open vessel polymerization. Such open vessel reactions, however, are not compatible with the moisture and oxygen sensitive transition metal catalyst. We have included these new data and related discussions into the revised manuscript accordingly.

1. The stability of NCAs in THF/DMF is not a particularly interesting addition to the manuscript. The data is not informative since the ppm water is not characterized or standardized. Solvents should be analyzed by a Karl Fischer titrator. The results reported for NCA stability in THF vs DMF probably don't indicate stability in these solvents, but rather, may just be a canary in the coal mine for wet DMF. Further, chloride ions are known initiators in DMF so NCA purity may also be a factor. The SI says anhydrous solvents were purchased and used without further purification, or were dried over MgSO₄, or distilled (for THF). The water content of 'anhydrous' solvents varies widely by manufacturer, by lot number, and in laboratory handling. Distillation is subject to the technique of the chemist and can result in THF of different ppm water, and MgSO₄ is only a crude method of drying. DMF is particularly challenging to dry. Further, depending on how the DMF was purified it may be contaminated with dimethylamine, which can initiate NCA polymerization. So overall, these data are not thoroughly examined for all factors and should be removed from the paper.

Response: We thank the reviewer for the comments that inspired us to have more detailed analysis on the solvent issue. As suggested by the reviewer, both THF and DMF were purchased from Sigma-Aldrich as anhydrous solvent and used directly in the glove box for NCA stability test. We followed the suggestion of the reviewer and carefully measured the water content of anhydrous DMF and THF using a Karl Fisher titrator. The actual water content of our DMF and THF was found to be 80 ppm and 46 ppm respectively that are slightly higher than the numbers, 20 ppm and 10 ppm water content respectively for DMF and THF, in the quality control report from Sigma-Aldrich. We agree with the reviewer that moisture in the solvent can contribute to the loss of NCA and the difference of NCA stability in THF vs. DMF. The reviewer also pointed out an important possibility of chloride ion effect on observed NCA stability issue in different solvent because chloride ions are known initiators in DMF. To examine if chloride ions can be a factor, we characterized our NCA using Inductively Coupled Plasma Mass Spectrometry (ICP-

MS). The content of chloride ions were found to be 1.5 ppm and 5.5 ppm respectively for BLG NCA and Boc-D,L-Lys NCA used in polymerization studies. Our NCA monomers have quite low chloride content compared with NCA monomers that were reported in literatures to polymerize successfully (Polymer Chemistry 2014, 52, 1385–1391; Polymer Chemistry 2016, 54, 311–315). So, impurity of chloride ions in NCA is likely not a major concern for observed stability issue of NCA in DMF.

We agree with the reviewer's comments that "The water content of 'anhydrous' solvents varies widely by manufacturer, by lot number, and in laboratory handling. Distillation is subject to the technique of the chemist and can result in THF of different ppm water. DMF is particularly challenging to dry ...". These are exactly part of the considerations for us to develop the LiHMDS initiated NCA polymerization that has fast reaction speed to suppress side reactions and influence of solvent like variation in water content. As demonstrated in our manuscript, LiHMDS initiated NCA polymerization proceeded successfully in open vessels outside of the glovebox even using standard AR grade THF without drying treatment (water content at 500 ppm as carefully measured using a Karl Fisher titrator). We think above advantages of LiHMDS initiated NCA polymerization will be interesting to researchers who study NCA chemistry or the functions & applications of NCA derived polypeptides. More and more researchers get involved in studies of NCA derived polypeptides and many of these researchers may have backgrounds in biology, materials, drug delivery, and tissue engineering, rather than chemistry. So, it is especially challenging for those researchers to run NCA polymerizations that have strict requirement in dry conditions or need special skills in operation and solvent treatment. As mentioned by the reviewer, "different labs will have variable levels of facilities, skills and care in handling anhydrous solvent and solvent treatment. In addition, water contents of anhydrous solvents may vary in different solvent providers and different lots from the same provider." Therefore, our LiHMDS initiated NCA polymerization provides an easy handle and fast synthetic strategy for researchers who will can have little concern on solvent treatment and dry working environment. We included these results and discussions into the revised manuscript.

Nonetheless, we agree with the reviewer that the stability of NCAs in THF/DMF is not a particularly interesting addition to the manuscript and the reason for the stability difference can be complicated. So, we took the reviewer's suggestion and removed related information in Figure 1a, 1b of our original manuscript from the main text. Since this information could remind

researchers pay attention to solvents if they have problems with their NCA polymerization using conventional initiator like primary amine, we moved our observation to the supporting information.

2. The devil may also be in the details of commercial LiHMDS. There are two processes used to prepare the reagent, via n-BuLi or Li metal, and LiHMDS from different suppliers may be prepared by different methods and polymerization results may vary. LiHMDS quality has troubled Pfizer process chemists. The authors should verify that LiHMDS from different sources or different lots give similar results.

[dx.doi.org/10.1021/op4002356](https://doi.org/10.1021/op4002356)

[dx.doi.org/10.1021/op400236r](https://doi.org/10.1021/op400236r)

[dx.doi.org/10.1021/op400237j](https://doi.org/10.1021/op400237j)

Response: We thank the reviewer for the important question about possible variation in commercial LiHMDS. We followed the suggestion from the reviewer and examined polymerization of BLG NCA using LiHMDS from different sources, different lots and different synthetic methods. We tested four different lots of LiHMDS from Sigma-Aldrich (SHBH2360V, SHBH9931, SHBH9029, SHBH8213). We also tested LiHMDS prepared from Li metal (purchased from HWRK Chem) and LiHMDS prepared from n-BuLi (purchased from Adamas). As summarized in the Supplemental Figure 2 of the revised manuscript, all six LiHMDS gave similar results in generating 350 mer poly-BLG. All the data, results including the full scale GPC traces, and discussions were included into the revised manuscript.

3. LiHMDS polymerization of Boc-L-Lys NCA is reported to take 3.5-10x longer to complete than BLG-NCA. It is strange that the kinetics of the two are so different and the authors should investigate this further. The kinetics put into question the utility of the initiator for construction of the proposed polypeptide libraries. If the kinetics are so different for various NCAs, the initiator would not be useful for statistical copolymers. The antimicrobial polypeptides reported in Fig 4 are more likely to be tapered polymers than statistical mixtures as shown.

Response: We thank the reviewer for comments on reaction kinetics of different NCA monomers. In fact, we used a racemic mixture of Boc-D,L-Lys NCA rather than homochiral Boc-L-Lys NCA in our original manuscript. The reviewer's question reminds us that using "Lys-NCA" as the abbreviation for racemic Boc-D,L-Lys NCA in our original manuscript can be misleading. Therefore, we changed the name of this racemic NCA as Boc-D,L-Lys NCA throughout the revised manuscript to avoid misunderstanding.

In response to the reviewer's question about reaction kinetics of different NCA, we did extra experiments and obtained the reaction rate for homochiral Boc-L-Lys NCA. As summarized in Figure 2f of the revised manuscript, the reaction rate of homochiral Boc-L-Lys NCA at 1.57 h^{-1} is similar to the reaction rate of BLG-NCA at 1.69 h^{-1} . So, LiHMDS can be used to generate statistical copolymers theoretically. In Figure 1 of our original manuscript, the data was presented for the NCA conversion of a homochiral BLG NCA and a racemic mixture of Boc-D,L-Lys NCA. Our purpose is to compare different NCA polymerization methods using one normal case (the BLG NCA) and another difficult case (the racemic Boc-D,L-Lys NCA). All the new data and discussions were included into the revised manuscript.

We chose a racemic mixture of Boc-D,L-Lys NCA to compose antimicrobial polypeptides with special reasons as described below, though copolymerization using homochiral Boc-L-Lys NCA is assumed to be more easy. One of the innate and prominent shortcomings of host defense peptide (HDP) is their low stability upon proteolysis. So, we used racemic Boc-D,L-Lys NCA rather than homochiral Boc-L-Lys NCA as the cationic subunit within the antimicrobial polypeptides to make the resulting polypeptide resistant to proteolysis. These amphiphilic copolymers were quickly synthesized to have narrow dispersity index ($\text{Đ}=1.16-1.17$) and incrementally increased ratios of the cationic subunit. The large difference in reaction rate between BLG NCA and Boc-D,L-Lys NCA (Figure 2f in the revised manuscript) encouraged us to explore NCA subunit composition of resulting polypeptides. Analyzing on subunit composition within polymer chains along the polymerization progress (or the NCA monomer conversion) indicated a higher content of BLG than Boc-D,L-Lys at the C-terminus before 30% assumption of the total NCA monomers. As a consequence, more Boc-D,L-Lys gets into the polymer chain than does BLG at the N-terminal after 80% consumption of total NCA monomers (Figure 5c in the revised manuscript). It is noteworthy that almost equal ratio of two subunits exists for majority of the polymer in middle of the polymer chain, even though there are huge

difference in reaction rate between BLG NCA and the racemic Boc-D,L-Lys NCA. This result also implies that theoretically statistical copolymers can be prepared in LiHMDS initiated NCA polymerization if NCA monomers with close reaction rate are used as the combination. Although the resulting polymers are not globally random copolymer, they are still valid as antimicrobial mimics of HDP because our previous study confirmed that some extent of intermixing of two subunits is enough to mimic HPD and have potent antimicrobial activities (*J. Am. Chem. Soc.* 2014, 136, 4410-4418). Our antimicrobial study results also supported this hypothesis as described in Figure 5d of the revised manuscript. All the data and discussions were included into the revised manuscript.

Items for correction or clarification:

1. What is the mobile phase for the HPLC used to determine stability of NCAs and conversion to polymer? If this is not run in anhydrous aprotic solvents, the data will be skewed by NCA hydrolysis and polymerization.

Response: We thank the reviewer for this important question. Reverse phase HPLC analysis on NCA stability and reaction progress used the combination of water (eluent A) and acetonitrile (eluent B) as the mobile phase. For isocratic elution, 90% B was used as the mobile phase to run for 6 min at a flow rate of 1 mL/min. For gradient elution, 60-100% B was used as the mobile to run for 16 min at a flow rate of 1 mL/min. BLG NCA were analyzed by integration of the absorbance peak at 254 nm. Boc-D,L-Lys NCA and Boc-L-Lys NCA were analyzed by integration of the absorbance peak at 210 nm. To examine if the water in the HPLC eluent can have significant effect on the result of analysis during the short time of HPLC elution, we did stability studies on NCA in 90% and 60% acetonitrile respectively. Since BLG NCA is much less stable than the Boc-D,L-Lys NCA, we examined if the HPLC condition is compatible with BLG NCA. According to HPLC result, the retention time for the BLG NCA is less than 3 min using 90% acetonitrile as the isocratic mobile phase, and the retention time for the BLG NCA is 4 min using 60-100% acetonitrile as the gradient mobile phase. The BLG NCA was dissolved in either 90% or 60% acetonitrile and kept for a series of chosen time. Then at different time point the NCA solution was injected into a HPLC using 100% acetonitrile as the mobile phase to calculate the remaining BLG NCA using an internal standard and a calibration curve

(Supplementary Figure 4 in the revised manuscript). Based on our studies, 99.8% and 98.9% of BLG NCA remains after the compound was kept for 4 mins in 90% and 60% acetonitrile respectively (Supplementary Figure 3 in the revised manuscript). The retention time for the BLG NCA is less than 3 min using 90% acetonitrile as the isocratic mobile phase, and the retention time for the BLG NCA is 4 min using 60-100% acetonitrile as the gradient mobile phase. That means the loss of NCA under our HPLC analysis condition is trivial and the data and result won't be skewed by possible NCA hydrolysis. All the data and discussions were included into the revised manuscript.

2. About 10 years ago IUPAC recommended discontinuing the use of the misleading phrase "Polydispersity index" and "PDI". Dispersity index should be used instead.

Response: We thank the reviewer for pointing out this. We followed the request and changed all "Polydispersity index" and "PDI" to "dispersity index" and the symbol "Đ" throughout the modified manuscript.

3. Lys-NCA should be called Boc-L-Lys-NCA if the protecting group for Glu NCA is noted. Naming should be standardized.

Response: We thank the reviewer for pointing out this. As suggested by the reviewer, we standardize all NCA in the modified manuscript. The Glu NCA is name as "BLG NCA". The N ϵ -tert-butyloxycarbonyl-D,L-lysine NCA named as "Lys-NCA" in our previous manuscript was changed to "Boc-D,L-Lys NCA" because it represents a racemic mixture as we used in our study. To address the reviewer's third general question above, in the revised manuscript we also studied the homochiral "Boc-L-Lys NCA" with its name following the suggestion by the reviewer. All other NCAs used in the revised manuscript were also named by following the same rule.

4. *Table 1 should have m:I ratios.*

Response: We have followed the request from the reviewer and added m:I ratios into the revised Table 1 and other Tables in our revised manuscript.

5. *Fig 1c, Boc-L-Lys NCA is missing stereochemistry.*

Response: We thank the reviewer for this comment. The “Lys-NCA” in Figure 1c of our original manuscript is actually a racemic mixture, but not a homochiral NCA. That is why we draw this NCA structure without stereochemistry. To avoid misleading and misunderstanding, we changed the name of this racemic “Lys-NCA” to “Boc-D,L-Lys NCA” in the modified manuscript to differentiate from the homochiral Boc-L-Lys NCA.

6. *The term ‘deprotonating agent’ could be better stated as a ‘base’.*

Response: We took the suggestions from the reviewer and switch the term “deprotonating agent” to “base” throughout the modified manuscript.

7. *GPC traces are truncated in the figures which could obscure information. The should be shown in full in the SI if there’s not room in the main text.*

Response: We thank the reviewer for this comment. The full GPC traces for polymers were added to the supplementary information in the revised manuscript.

8. *There are a considerable number of typos and convoluted language. The authors are advised to seek a native English speaker for editing service.*

Response: We thank the reviewer for this comment and suggestions to help us further improve the quality of our manuscript. We got generous help from Professor Samuel Gellman at UW-Madison to polish the English of the revised manuscript.

We hope that the revised manuscript will prove to be acceptable for publication in *Nature Communications*. Thank you for your assistance.

Sincerely,

Runhui Liu

Professor of Chemistry and Biomaterials

REVIEWERS' COMMENTS:

Reviewer #1 (Remarks to the Author):

The authors have addressed all the concerns I raised in the initial revision. For the open-flask operation, the authors have demonstrated that it can be used to synthesize a various of polymers with a high polymerization degree, the polymer chain is still active and can be further extended to prepare block copolymer and it can be used to polymerize a various of monomers. For the mechanism study, the authors prove the intermediate structure and support the mechanism.

In general, the authors have addressed all the concerns and I think it can be accepted in its current form.

Reviewer #2 (Remarks to the Author):

This revised manuscript addresses most of my previous concerns. All the questions have been adequately addressed in this revision. As a last minor correction:

Lines 117-118 « To answer whether LiHMDS-initiated NCA polymerization can provide polypeptides with defined secondary structures,... » This sentence should be supported by the following reference Polym. Chem. 2018, 9, 1517-1529.

I then strongly recommend this manuscript for publication in Nature Communications and I greatly congratulate all the authors for their contribution to the field and for their hard work.

Reviewer #3 (Remarks to the Author):

The authors have substantially revised the original manuscript and have adequately addressed the major concerns that were brought up in the original manuscript. Considering the ease of use of the new catalyst and the remarkable polymer molecular weights achieved with this system, the manuscript is recommended for publication.

One point of clarification is regarding the statement that transition metal catalysts limit biological applications. The catalysts can easily be removed from the polypeptides by simple precipitation and dialysis. The Deming lab has published ICP Mass Spec studies indicating efficient removal, and have extensively used polypeptides derived from this method for work in cells and animals. The truly exciting part of this manuscript is the tolerance of the catalyst to oxygen water and the resulting high molecular weight material.

We greatly thank all the reviewers for the favorable comments. All reviewers' comments and suggestions, throughout the review process, are very helpful, and the resulting changes have significantly improved the manuscript.

Our responses to reviewer comments are detailed below; for reference, the reviewer comments/questions are presented in *italics*, with our response following.

Reviewer 1

Comments: The authors have addressed all the concerns I raised in the initial revision. For the open-flask operation, the authors have demonstrated that it can be used to synthesize a various of polymers with a high polymerization degree, the polymer chain is still active and can be further extended to prepare block copolymer and it can be used to polymerize a various of monomers. For the mechanism study, the authors prove the intermediate structure and support the mechanism.

In general, the authors have addressed all the concerns and I think it can be accepted in its current form.

Response: We thank the reviewer for the favorable comments. We greatly appreciate all comments and suggestions from the reviewer to help us substantially improve our manuscript.

Reviewer 2

Comments: This revised manuscript addresses most of my previous concerns. All the questions have been adequately addressed in this revision. As a last minor correction: Lines 117-118 « To answer whether LiHMDS-initiated NCA polymerization can provide polypeptides with defined secondary structures,... » This sentence should be supported by the following reference Polym. Chem. 2018, 9, 1517-1529.

I then strongly recommend this manuscript for publication in Nature Communications and I greatly congratulate all the authors for their contribution to the field and for their hard work.

Response: We thank the reviewer for the favorable comments that encourages us to work even harder in the future. We also thank the reviewer for all previous comments and suggestions to help us greatly improve our manuscript. The paper suggested (Polym. Chem. 2018, 9, 1517-

1529) by the reviewer is a great reference for characterization and understanding of secondary structure of polypeptides. We have included this paper as ref 64 in the main text of our revised manuscript.

Reviewer 3

Comments: *The authors have substantially revised the original manuscript and have adequately addressed the major concerns that were brought up in the original manuscript. Considering the ease of use of the new catalyst and the remarkable polymer molecular weights achieved with this system, the manuscript is recommended for publication.*

One point of clarification is regarding the statement that transition metal catalysts limit biological applications. The catalysts can easily be removed from the polypeptides by simple precipitation and dialysis. The Deming lab has published ICP Mass Spec studies indicating efficient removal, and have extensively used polypeptides derived from this method for work in cells and animals. The truly exciting part of this manuscript is the tolerance of the catalyst to oxygen water and the resulting high molecular weight material.

Response: We thank the reviewer for the favorable comments and the special clarification on Prof. Deming's transition metal catalyst. In addition to this, our side-by-side comparison on our method with the transitions metal catalyzed method, as request by the reviewer, reinforces the excellence and advantages of the Ni-catalyst than primary initiators.

We greatly appreciate all comments and suggestions from the reviewer to help us substantially improve our manuscript.

We hope that the revised manuscript will prove to be acceptable for publication in *Nature Communications*. Thank you for your assistance.